# ASIC1a affects hypothalamic signaling and regulates the daily rhythm of body temperature in mice

Zhong Peng[1], Panos G. Ziros [2], Tomaz Martini[3,8], Xiao-Hui Liao[4], Ron Stoop[5], Samuel Refetoff[4,6,7], Urs Albrecht [3], Gerasimos P. Sykiotis[2] & Stephan Kellenberger [1✉]

The body temperature of mice is higher at night than during the day. We show here that global deletion of acid-sensing ion channel 1a (ASIC1a) results in lower body temperature during a part of the night. ASICs are pH sensors that modulate neuronal activity. The deletion of ASIC1a decreased the voluntary activity at night of mice that had access to a running wheel but did not affect their spontaneous activity. Daily rhythms of *thyrotropin-releasing hormone* mRNA in the hypothalamus and of *thyroid-stimulating hormone β* mRNA in the pituitary, and of *prolactin* mRNA in the hypothalamus and pituitary were suppressed in ASIC1a$^{-/-}$ mice. The serum thyroid hormone levels were however not significantly changed by ASIC1a deletion. Our findings indicate that ASIC1a regulates activity and signaling in the hypothalamus and pituitary. This likely leads to the observed changes in body temperature by affecting the metabolism or energy expenditure.

[1] Department of Biomedical Sciences, University of Lausanne, Lausanne, Switzerland. [2] Service of Endocrinology, Diabetology and Metabolism, Lausanne University Hospital and University of Lausanne, Lausanne, Switzerland. [3] Department of Biology/Unit of Biochemistry, Faculty of Sciences, University of Fribourg, Fribourg, Switzerland. [4] Department of Medicine, The University of Chicago, Chicago, IL, USA. [5] Center for Psychiatric Neurosciences, Hôpital de Cery, Lausanne University Hospital, Lausanne, Switzerland. [6] Department of Pediatrics, The University of Chicago, Chicago, IL, USA. [7] Committee on Genetics, The University of Chicago, Chicago, IL, USA. [8]Present address: Institute of Bioengineering, School of Life Sciences, École Polytechnique Fédérale de Lausanne, Lausanne, Switzerland. ✉email: stephan.kellenberger@unil.ch

In mammals, the body temperature is actively controlled and follows a circadian rhythm. The hypothalamus senses the body temperature and controls the balance of heat production and loss. The latter is achieved by adjusting food intake, metabolism, and energy expenditure through hormonal and neuronal mechanisms. Endocrine regulation of heat production can involve the hypothalamic-pituitary-thyroid (HPT) axis. The HPT axis is under the control of the suprachiasmatic nucleus (SCN)[1]. The circadian timing system comprises a central pacemaker located in the SCN of the hypothalamus and peripheral oscillators (tissue clocks)[2]. External stimuli such as light influence the SCN via the retinohypothalamic tract from the retina to adjust circadian rhythms to the daily light-dark cycle.

In this study, we examined the impact of the acid-sensing ion channel (ASIC) 1a on daily rhythms, and notably on body temperature regulation. ASICs are non-voltage-gated $Na^+$ channels of the nervous system that are activated by extracellular acidification[3,4]. ASICs of the CNS are homo- or heterotrimers comprised of ASIC1a, ASIC2a, and/or ASIC2b[5]. ASIC1a confers high pH sensitivity to brain ASICs[6]. It is distributed throughout the central and peripheral nervous systems, participating in fear conditioning[7] and synaptic transmission and plasticity[6,8]. Dysfunction of ASIC1a is associated with the development of diverse neurological diseases, including neurodegeneration after ischemic stroke[9] and neurodegenerative diseases[10]. The activation of ASICs induces action potentials (APs) and leads to the excitation of neurons[11,12]. The $Ca^{2+}$ influx induced by the activation of ASIC1a leads to the activation of diverse signaling pathways[13,14]. We show here that the expression and function of ASIC1a in the mouse hypothalamus follow a daily cycle, and that global deletion of ASIC1a changes the body temperature during the night. In ASIC1a$^{-/-}$ mice, signaling in the hypothalamus and the pituitary was altered, but serum thyroid hormone levels were unchanged. The decrease in body temperature of ASIC1a$^{-/-}$ mice is therefore likely mediated by changes in food intake, metabolism, and/or energy expenditure.

## Results

**Daily rhythm of ASIC1a expression in the hypothalamus.** To determine whether ASIC1a expression in the brain follows a daily rhythm, the hypothalamus, SCN, and hippocampus were isolated at 4 h intervals from 3-month-old WT male mice entrained to a standard 12 h light:12 h dark (LD) cycle. Western blot analysis showed an increase in the abundance of ASIC1a protein at Zeitgeber time (ZT) 8–16 in the hypothalamus (Fig. 1a), a decrease at ZT8-12 in the SCN (Fig. 1b), and a small decrease at ZT4 in the hippocampus (Supplementary Fig. 1a). The *Asic1a* mRNA level in the hypothalamus was higher at ZT8-16 than at ZT20 (Fig. 1c), and the level in the SCN was higher at ZT0 than at ZT16 (Fig. 1c). In hypothalamic tissue from which the SCN had been removed, ASIC1a expression peaked at ZT8-12 (Supplementary Fig. 1b).

Because the expression and function of ASIC1a in the hypothalamus follows a daily cycle, we tested whether ASIC1a affects the circadian clock. In the hypothalamus of WT mice under a standard LD cycle, the abundance of the brain and muscle Arnt-like protein (BMAL1), which is crucial for normal molecular clock function, showed a daily rhythm in hypothalamus and hippocampus; its cycle was shifted between total hypothalamus and SCN (Fig. 1a, b, Supplementary Fig. 1a). A rhythmic pattern of *Bmal1* mRNA expression in the hypothalamus and SCN was also observed (Fig. 1d). The daily rhythm of BMAL1 protein abundance in the hypothalamus was shifted by the global deletion of ASIC1a compared to WT mice (Supplementary Fig. 1c). This effect, albeit subtle, may indicate a regulatory action of ASIC1a on the circadian clock or vice versa.

**Daily activity patterns of hypothalamic neurosecretory neurons are modulated by ASIC1a.** We asked whether ASIC function in hypothalamic neurons also follows a daily rhythm. A first series of experiments was carried out in cultured hypothalamic neurons whose cycle had not been synchronized. The neurons from which recordings had been made were grouped into 4 types, according to their morphology. The pH6.6-induced ASIC current density— measured as ratio of the peak current amplitude of the transient ASIC current/cell capacitance—showed a daily rhythm in type 2 (Supplementary Fig. 2a), and a tendency towards a daily rhythm in type 3 ($p = 0.086$ between 12 h and 18 h). The pH6.6-induced ASIC current density was higher in hypothalamic neurons than in neurons from mouse cortex or hippocampus (Supplementary Fig. 2b). To distinguish the measured neurons into neurosecretory and non-neurosecretory ones, published electrophysiology protocols (Supplementary Fig. 2c)[15,16] were applied to current-clamped neurons of type 1–3, and neurons were classified according to their voltage response. This analysis classified 56% of type 1 neurons as neurosecretory neurons, while this proportion was 14% in type 2 and 25% in type 3 neurons (Fig. 2a). Data shown in Fig. 2b–e, are from type 1 neurosecretory neurons that had been exposed for 30 min to 1 μM dexamethasone to synchronize their daily rhythm[17]. After synchronization treatment, the acid-induced ASIC current density showed a daily rhythm with a maximal amplitude at ZT12 (Fig. 2b). Exposure to pH6.6-induced APs in these current-clamped neurons due to ASIC-mediated depolarization (Fig. 2c). The number of acid-induced APs showed a daily rhythm, with a maximum at ZT0 (Fig. 2d), thus at a ZT at which the ASIC peak current expression was not maximal. The area under the curve (AUC) of the pH6.6-induced depolarization in these experiments, which is a measure of the pH6.6-induced ASIC activity, was maximal at ZT12 (Fig. 2e), thus at the same ZT as maximal ASIC peak currents. Strong depolarizations can limit the number of APs by a mechanism that resembles a depolarization block[12,18], explaining the shifted maxima between AP number and ASIC current density. In physiological situations, acidifications are likely smaller than the change from pH7.4 to 6.6 that was used in these experiments, and therefore, the maximal ASIC-mediated AP signaling would likely be observed at the time of maximal ASIC expression. The observations in Fig. 2b–e are consistent with data obtained without synchronization treatment (Supplementary Fig. 2d, e; note that the timing is indicated as daytime, not ZT time). Thus, the function of ASICs in the investigated embryonic neurosecretory hypothalamic neurons has a daily rhythm and affects neuronal signaling.

**ASIC1a affects neuronal activity in the paraventricular nucleus of the hypothalamus (PVH).** ASIC1a mRNA is highly expressed in the hypothalamus[19,20]. In the PVH, the neurosecretory neurons are concentrated in the medial parvocellular division (mpd) of the PVH[15] (Fig. 2f). Rhythmic activity of the hypothalamus is controlled by the SCN. The spontaneous firing rate (SFR) of SCN neurons in hypothalamic slices shows a synchronous daily rhythm[21]. We recorded single-unit activity in the mpd from acute coronal hypothalamus slices with the loose patch technique, to measure the SFR of mpd neurons during day- and nighttime. We observed higher activity at night in WT but not ASIC1a$^{-/-}$ neurons (Fig. 2g, h). The SFR measured at night was higher in WT than ASIC1a$^{-/-}$ neurons, whereas there was no significant difference at daytime (Fig. 2h), suggesting that the deletion of ASIC1a affected only the nocturnal activity of these predominantly neurosecretory neurons.

**ASIC1a regulates the daily rhythm of the body temperature.** Under a LD cycle, both WT and ASIC1a$^{-/-}$ mice showed daily rhythms in body temperature as expected[22], with higher

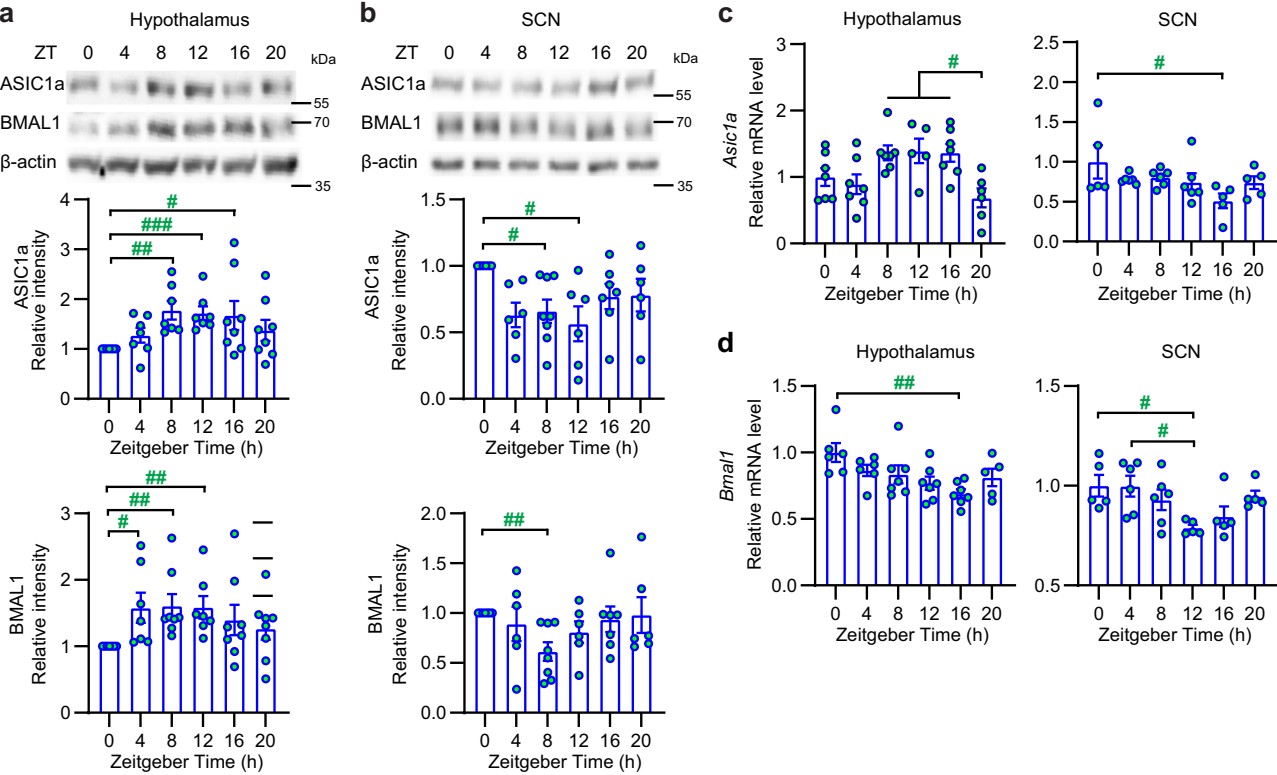

**Fig. 1 Daily pattern of ASIC1a and BMAL1 expression in mouse brain. a, b** The biochemical experiments were carried out with tissues of male mice, and proteins were separated by SDS-PAGE. Representative western blots of ASIC1a, BMAL1, and β-actin and quantitative analysis are shown for each protein at the indicated ZT in the hypothalamus (**a**) and SCN (**b**). β-actin was used as a loading control for the total protein. For the quantification of ASIC1a and BMAL1 protein expression, the intensity of each protein band is normalized to the corresponding band at ZT0 in each independent experiment; samples are from $n = 6$–8 animals per condition. **c, d** RT-qPCR analysis in mouse hypothalamus and SCN of *Asic1a* mRNA expression (**c**) and *Bmal1* mRNA expression (**d**). Results for each mouse are presented as relative expression normalized to the mean of the ZT0 group, $n = 5$–7 animals per condition. #$p < 0.05$; ##$p < 0.01$; ###$p < 0.001$; comparison of log2-transformed data between time points, by one-way ANOVA and Tukey post hoc test and in **a**, **b** to the ZT0 value by one-sample *t* test (Methods). Error bars indicate SEM.

temperature during the active period (Fig. 3a). The body temperature was lower in ASIC1a$^{-/-}$ than WT mice at ZT17 and ZT18. Spontaneous activity, measured by the implanted sensor, was not significantly different between WT and ASIC1a$^{-/-}$ mice (Fig. 3b), suggesting that the decreased body temperature in ASIC1a$^{-/-}$ mice was not caused by a reduced spontaneous activity. The thyroid hormones triiodothyronine (T3) and thyroxine (T4) are the main regulators of the basal metabolic rate and thermogenesis; their formation and release from the thyroid is stimulated by the secretion of thyrotropin-releasing hormone (TRH) from the PVH, which upregulates the synthesis and secretion of thyroid-stimulating hormone (TSH) by the pituitary[23]. The secretion of TRH and the circadian regulation of the HPT axis are controlled by neurosecretory neurons in the hypothalamus. The *Trh* mRNA expression in the WT hypothalamus showed a daily rhythm with a peak at ZT12 (Fig. 3c), while such a rhythm was absent in ASIC1a$^{-/-}$ mice (Fig. 3d). The *Trh* mRNA in the PVH—from where TRH that regulates T4 and T3 originates—was increased at ZT13 relative to ZT1 in WT but not in ASIC1a$^{-/-}$ mice and was higher in WT than in ASIC1a$^{-/-}$ at ZT13 (Fig. 3e). The two time points of ZT1 and ZT13 for this mRNA expression measurement and the analysis of other genes were chosen since they correspond to the time point briefly after light switching.

**ASIC1a affects gene expression in the hypothalamus.** To better understand how ASIC1a affects signaling in the hypothalamus,

we performed a global transcriptome deep sequencing of 3-month-old male WT and ASIC1a$^{-/-}$ mice that had been kept under a standard LD cycle. The hypothalamus was isolated at ZT1 or ZT13, and RNA levels were quantified by next-generation sequencing (RNA-seq, *Methods*). Only 18 genes showed significant differences in their hypothalamic expression levels between ASIC1a$^{-/-}$ and WT mice (Fig. 4a). At ZT1, one gene (Keratin 90 pseudogene, *Krt90*) was downregulated, and one gene (*Cers5*) was upregulated in ASIC1a$^{-/-}$ hypothalamus (Fig. 4a, b). At ZT13, 3 genes (*Krt90*, *Prl*, and *Gm49980*) were downregulated and 13 genes were upregulated in the ASIC1a$^{-/-}$ hypothalamus (Fig. 4a, b). *Prl*, prolactin, is a pituitary hormone that has also been detected in the amygdala, the preoptic area of the hypothalamus and in the olfactory bulb[24]. Other genes regulated by ASIC1a are signaling proteins, including a GTPase (*Arl4d*), a kinase (*Sgk1*), transcription factors (*Btg2, Egr1-3*), and nuclear receptors (*Nr4a1,3*). They are all negatively regulated by activation of Akt or mTOR[25]. The *c-fos, fosb*, and *Junb* genes are early response genes (ERG) that are activated transiently and rapidly in response to a wide variety of cellular stimuli; they are also regulated by Akt or mTOR[26]. *1700016P03Rik* encodes the microRNAs 132 and 212[27], and *Gm49980* is a long intergenic non-coding RNA gene.

The expression of these genes, except for the non-coding genes and *Krt90*, was tested by RT-qPCR in the same mouse hypothalamus RNA samples. The expression of *Prl* was increased at ZT13 relative to ZT1 in WT but not ASIC1a$^{-/-}$ mice and was higher in the WT than in ASIC1a$^{-/-}$ mice at ZT13 (Fig. 4c). The

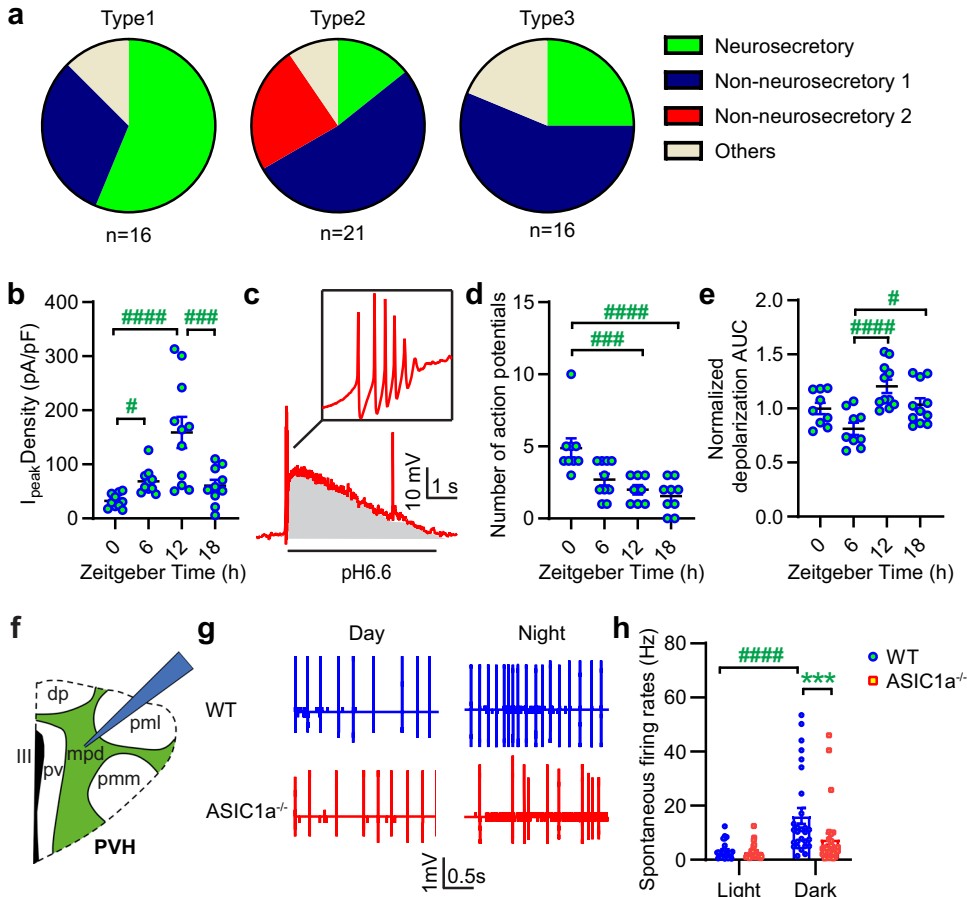

**Fig. 2 ASIC1a regulates the hypothalamic signaling activity. a** Cultured hypothalamus neurons were classified into types 1–3 as indicated in Supplementary Fig. 2a. Neurons were exposed to current protocols, indicated in Supplementary Fig. 2c, to determine the proportion of secretory and non-secretory neurons. **b–e** data obtained after synchronization by dexamethasone of cultured mouse neurosecretory Type 1 hypothalamus neurons. **b** Peak current densities of pH6.6-induced currents at the indicated ZT, measured by voltage-clamp at −60 mV; $n = 10$-11 cells per condition. **c–e** Current-clamp measurements; **c** Representative voltage trace of pH6.6-induced depolarization. **d** Number of pH6.6-induced action potentials (APs) at the indicated ZT; $n = 8$-10 cells per condition. **e** Normalized AUC of the depolarization (voltage · time, gray area in **c**) of pH6.6-induced depolarization at the indicated ZT; $n = 8$-10 cells per condition. **f–h** Extracellular recordings, from mouse brain slices. **f** Scheme of the coronal PVH slice map, from which recordings in loose-patch mode were done on mpd neurons. pv, periventricular parvocellular; dp, dorsal parvocellular; pml, posterior magnocellular lateral; pmm, posterior magnocellular medial; III, third ventricle. **g** Representative traces recorded during the day or night as indicated, from WT (blue) and ASIC1a$^{-/-}$ (red) brain slices. **h** Mpd neuron SFR from WT and ASIC1a$^{-/-}$ mice. Cells not firing APs were not included in this analysis; $n = 26$-30 cells from four mice in each condition. Statistical analysis was done on log2-transformed data for **b**, **e**, **h**. One-way ANOVA test and Tukey post hoc test (**b**, **e**), Kruskal–Wallis and Dunn's post-hoc test of non-transformed data (**d**); two-way ANOVA and Tukey post hoc test (**h**). $\#p < 0.05$; $\#\#\#p < 0.001$; $\#\#\#\#p < 0.0001$ (**b–e**, **h**), comparison between day and night or different ZT for the same genotype. ***$p < 0.01$; WT compared to ASIC1a$^{-/-}$ at the corresponding period (**h**). Error bars indicate SEM.

indicated upregulation of *Cers5* in ASIC1a$^{-/-}$ was not confirmed with RT-qPCR (Fig. 4c). The only significant difference between WT and ASIC1a$^{-/-}$ mice at ZT1 was the higher expression of *Nr4a3* in WT hypothalamus (Fig. 4e). The higher expression in ASIC1a$^{-/-}$ relative to WT at ZT13 was confirmed for *Arl4d, Sgk1, Egr1, Egr3,* and *Nr4a1*. The RT-qPCR analysis also showed that the hypothalamic expression of several of these genes was increased at ZT13 relative to ZT1 in ASIC1a$^{-/-}$ mice (*Arl4d, Sgk1, Btg2, Egr1, Egr3, Nr4a1,* and *Nr4a3*), while such an increase was seen in WT hypothalamus only for *Nr4a3* (Fig. 4d, e). The expression of *Fosb* was not different between ZT1 and ZT13 in WT (Supplementary Fig. 3a). In ASIC1a$^{-/-}$ mice, its level was lower at ZT1 and higher at ZT13 than WT, and it was increased at ZT13 over ZT1. A similar pattern was observed for *c-fos*. For *Junb* in contrast, there was an increase at ZT13 over ZT1 in WT, but not ASIC1a$^{-/-}$ mice. The absence of *Asic1a* mRNA in ASIC1a$^{-/-}$ animals was confirmed (Supplementary Fig. 3b).

Although the RNA-seq did not show any difference in *Trh* levels (Fig. 4a, b), the RT-qPCR analysis showed a diurnal rhythm of *Trh* mRNA expression in the hypothalamus and the PVH, which was disrupted by the deletion of ASIC1a (Fig. 3c–e). The hypothalamic expression of the pituitary hormone *thyroid-stimulating hormone β subunit (Tshb)* was not different between WT and ASIC1a$^{-/-}$ mice, nor between the two time points (Supplementary Fig. 3c).

**The daily expression rhythms of *Trh* and *Prl* are regulated by ASIC1a, likely via the Akt-mTOR pathway.** Since several genes that were upregulated at ZT13 in the hypothalamus of ASIC1a$^{-/-}$ mice (Fig. 4) are known to be negatively regulated by Akt and/or mTOR signaling, we tested next whether Akt and mTOR signaling was affected by the deletion of ASIC1a. *Trh* and *Prl* expression was previously shown to depend on the activity of the Akt-mTOR pathway[28,29]. Since global deletion of ASIC1a

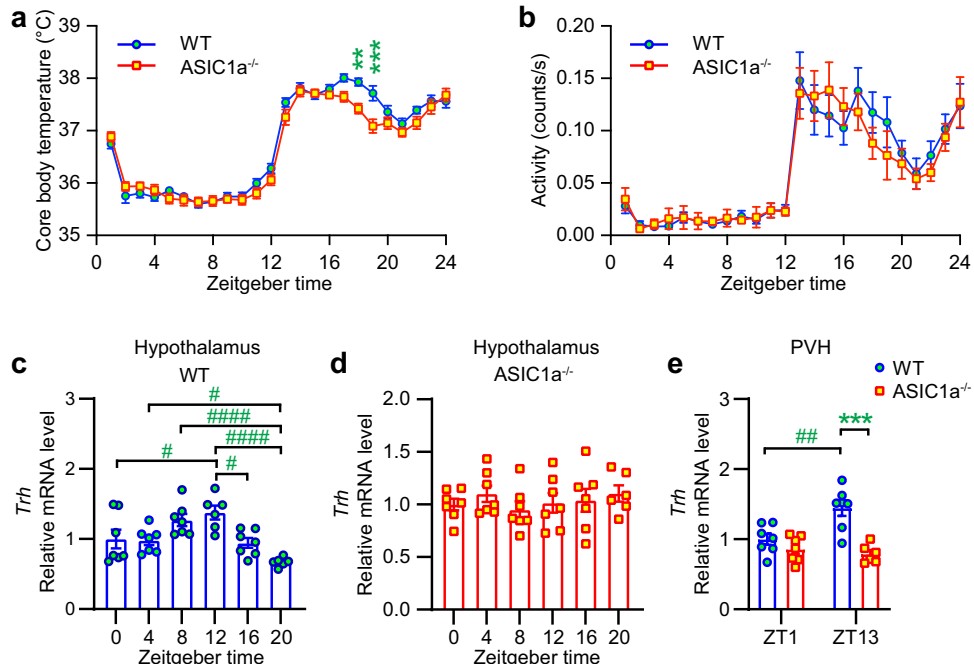

**Fig. 3 Global ASIC1a deletion affects body temperature and thyrotropin-releasing hormone mRNA levels in the hypothalamus. a, b** Mice were kept under a 12 h light:12 h dark (LD) cycle. Body temperature and locomotor activity of mice was measured with an implanted sensor. **a** Core body temperature rhythm for 24 h under LD cycle, $n = 10$ WT and 11 ASIC1a$^{-/-}$ mice. **b** Daily profile of locomotor activity in LD cycle. Activity counts are expressed as counts/s; $n = 10$ WT and 11 ASIC1a$^{-/-}$ mice. **c-d** RT-qPCR analysis of the daily profile of *Trh* expression in hypothalamus of WT (**c**) and ASIC1a$^{-/-}$ mice (**d**). Results for each mouse are presented as relative expression normalized to the mean at ZT0; $n = 6$-7 animals per condition. **e** *Trh* mRNA expression in the PVH at ZT1 and ZT13; normalization to the mean WT value at ZT1; tissue from $n = 5$-7 animals per condition. Statistical analysis was done on absolute values (**a, b**) or log2-transformed data in **c-e**. **$p < 0.01$; ***$p < 0.001$, different between genotypes, by two-way ANOVA, repeated measures, Sidak post hoc test (**a, b**). #$p < 0.05$; ##$p < 0.01$; ####$p < 0.0001$; comparison between time points; ***$p < 0.0001$, between genotypes (**c-e**); one-way ANOVA and Tukey post hoc test (**c, d**) and two-way ANOVA and Tukey post hoc test (**e**). Error bars indicate SEM.

altered the expression of *Trh* and *Prl* in the hypothalamus (Figs. 3c–e, 4c), we investigated possible involvement of the Akt-mTOR pathway in the regulation of the daily expression of *Trh* and *Prl* by ASICs. The protein abundance of Akt and mTOR showed no daily rhythm in either WT or ASIC1a$^{-/-}$ mice, with no significant difference between the two genotypes either (Fig. 5a and Supplementary Fig. 4a), consistent with the mRNA analysis (Supplementary Fig. 4b). Akt and mTOR are activated by phosphorylation induced by upstream signaling. The relative abundance of phosphorylated Akt (p-Akt), measured as p-Akt/Akt ratio, displayed in WT hypothalamus an increase at ZT12 relative to ZT0; in ASIC1a$^{-/-}$ hypothalamus, this ratio was higher at ZT0 than at ZT8 (Fig. 5a). The overall hypothalamic p-Akt/Akt ratio was slightly increased upon deletion of ASIC1a$^{-/-}$ ($p = 0.043$, Student's $t$-test). The p-mTOR/mTOR ratio showed a daily rhythm in both WT and ASIC1a$^{-/-}$ hypothalamus, with highest values at night in WT, and at ZT8 in ASIC1a$^{-/-}$ mice. The overall hypothalamic p-mTOR/mTOR ratio was decreased by ~50% upon deletion of ASIC1a$^{-/-}$ (Fig. 5a; $p < 0.0001$, Student's $t$-test), consistent with the observed higher expression in ASIC1a$^{-/-}$ hypothalamus of genes known to be negatively regulated by mTOR.

To test whether the observed changes in Akt and mTOR activation are due to altered ASIC expression or activity, the effects of the Akt inhibitor triciribine, the mTOR inhibitor rapamycin, the ASIC1a agonist Texas coral snake toxin (MitTx) and the ASIC1a inhibitor psalmotoxin1 (PcTx1) were measured. This set of experiments was carried out in cultured neurons of the cortex, which express ASIC currents[6,30]. When primary cultures of WT and ASIC1a$^{-/-}$ cortical neurons were incubated with the indicated inhibitors or agonists for 2 h, ASIC1a abundance was not changed (Supplementary Fig. 4c). Triciribine lowered

the p-Akt/Akt and p-mTOR/mTOR ratios or showed a tendency towards such a decrease, while rapamycin decreased or showed a tendency of a decrease of the p-mTOR/mTOR ratio in WT and ASIC1a$^{-/-}$ neurons (Supplementary Fig. 4c). Both ratios were increased with MitTx in WT but not ASIC1a$^{-/-}$ neurons, consistent with an activation of the Akt-mTOR pathway by ASIC1a; they remained however unaffected by exposure to PcTx1. Cultured cortical neurons were exposed to the drugs during 2 h, which is a different condition from the experiments in which Akt/mTOR signaling was compared between hypothalamus tissue of WT and ASIC1a$^{-/-}$ (Fig. 5a). This may explain the absence of an effect of PcTx1 exposure. The observed opposite change of Akt and mTOR activity in hypothalamus tissue by ASIC1a$^{-/-}$ deletion is incompatible with a unique signaling along the Akt-mTOR pathway and suggests that other interacting signaling pathways are also likely affected.

To test whether the activity of ASIC1a regulates the expression of *Prl* and *Trh* in hypothalamic neurons, the mRNA expression of *Prl* and *Trh* was measured in primary cultures of WT and ASIC1a$^{-/-}$ hypothalamic neurons after a 2-h incubation with these compounds. MitTx increased *Trh* and *Prl* expression in WT but not ASIC1a$^{-/-}$ neurons, whereas PcTx1 decreased *Trh* but not *Prl* in WT neurons (Fig. 5b, c). The expression of *Trh* was decreased in WT and ASIC1a$^{-/-}$ neurons after triciribine and rapamycin treatment. The *Prl* expression was decreased by rapamycin but not by triciribine in WT neurons. Triciribine increased the expression levels of *Arl4d*, *Sgk1*, *Egr1* and *Nr4a1* in WT and of *Nr4a1* in ASIC1a$^{-/-}$ hypothalamic neurons (Supplementary Fig. 4d), while rapamycin increased only the expression levels of *Sgk1* in WT neurons. Inhibition of ASIC1a by PcTx1 increased the mRNA levels of *Sgk1*, *Egr1*, *Egr3* and *Nr4a1*

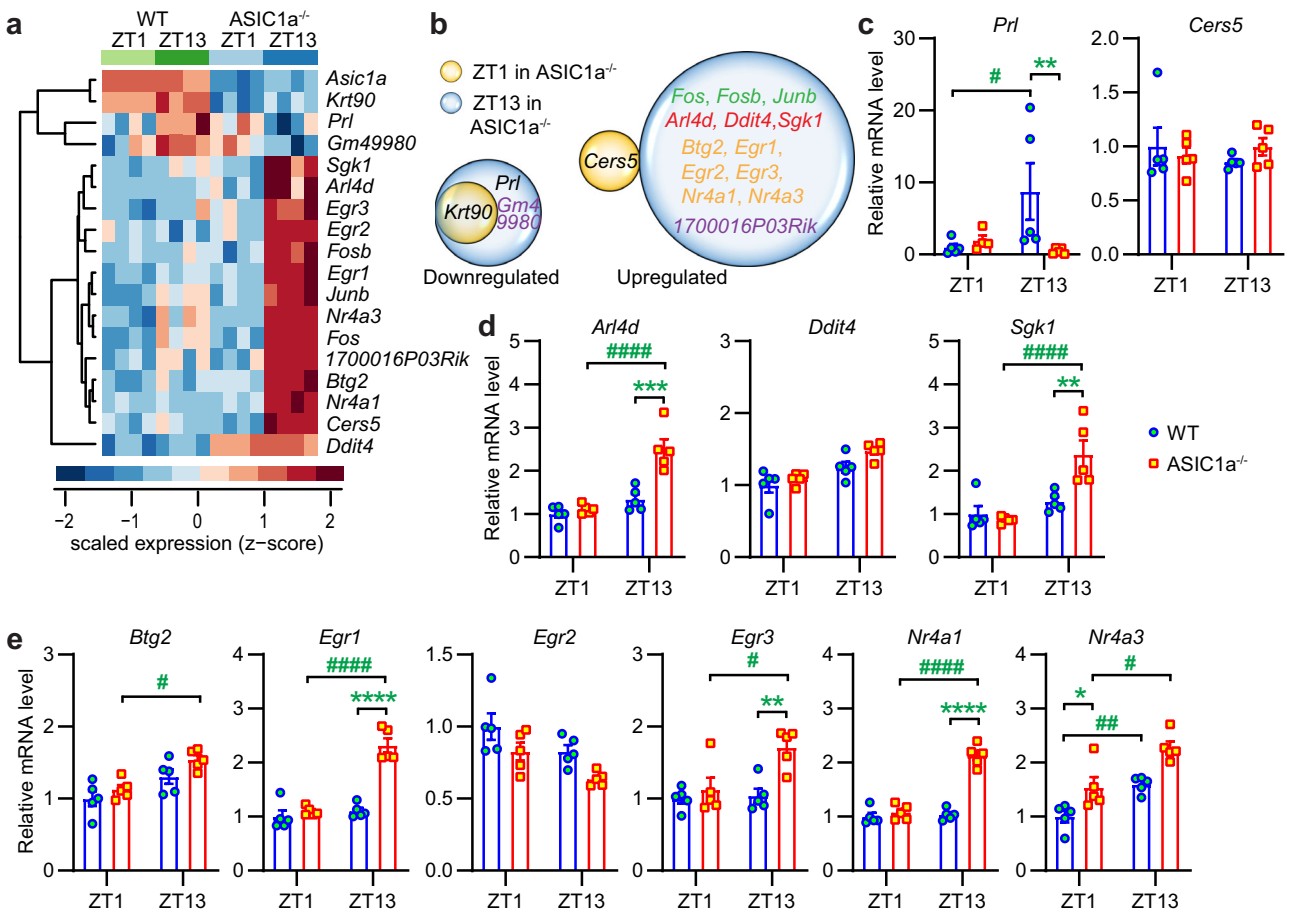

**Fig. 4 Effects of ASIC1a on the hypothalamus transcriptome. a** Heat map representing significantly altered transcripts (*p* < 0.005) in hypothalamus across the genotypes. Samples are in columns and genes are in rows; *n* = 4 animals per condition. High expression is displayed in red (*z* score >1) while low expression in blue (*z* score <1). **b** Venn diagrams comparing the expression-changed genes, based on the RNA-seq data sets. **c–e** RT-qPCR analysis to identify the gene expression change of functional (**c**), kinase and signaling protein (**d**), and transcription factor and nuclear receptor genes (**e**). Results for each mouse are presented as relative expression, normalized to the mean of the WT at ZT1; *n* = 4–5 animals per condition. *\*p* < 0.05; *\*\*p* < 0.01; *\*\*\*p* < 0.001; *\*\*\*\*p* < 0.0001; WT compared to ASIC1a⁻/⁻ at the corresponding ZT; #*p* < 0.05; ##*p* < 0.01; ####*p* < 0.0001; comparison for a given genotype between the ZT conditions; two-way ANOVA test and Tukey post hoc test, done on log2-transformed data. Error bars indicate SEM.

in WT but not ASIC1a⁻/⁻ neurons, consistent with a negative regulation by ASIC1a; however, treatment with the ASIC agonist MitTx did not downregulate the mRNA expression of these genes.

Activation of ASIC1a is known to allow Ca²⁺ influx and to activate PI3K, which in turn activates the Akt-mTOR pathway[31]. Activation of the Akt-mTOR pathway upregulates the cAMP response element-binding protein (CREB)[32,33], which is known to be involved in the transcriptional activation of *Prl* and *Trh*[34,35]. To further investigate the signaling pathway connecting ASIC1a to the expression of *Trh*, we determined whether pre-incubation with the following compounds affected *Trh* expression after MitTx exposure in cultured hypothalamus neurons: BAPTA-AM (a cell-permeable Ca²⁺ chelator), KG-501 (an inhibitor of p-CREB), triciribine, PcTx1 and rapamycin. All inhibitors decreased the expression of *Trh* after MitTx exposure in WT neurons (Fig. 5d), suggesting that the MitTx-induced expression level of *Trh* depends, at least in part, on intracellular Ca²⁺, the Akt-mTOR pathway, and p-CREB.

**Lower voluntary locomotor activity in ASIC1a⁻/⁻ mice.** To test whether ASIC1a may be involved in the control of diurnal rhythms, we studied locomotor activity rhythms of WT and ASIC1a⁻/⁻ mice by assessing wheel running. When exposed to

an LD cycle, locomotor activity in the dark period was lower in ASIC1a⁻/⁻ mice compared to WT mice (Fig. 6a, b), indicating that ASIC1a regulates activity rhythms. The free-running period, determined under constant darkness (DD) conditions, was slightly but significantly longer in ASIC1a⁻/⁻ mice compared to WTs mice (Fig. 6a, c). These data suggest an effect of ASIC1a on the molecular clock or modulation of molecular clock outputs. However, short light pulses at ZT14 or ZT22 immediately after releasing the mice to constant darkness (according to the Aschoff type 2 protocol[36]) shifted the cycle in the same way in WT and ASIC1a⁻/⁻ mice (Fig. 6d), indicating that ASIC1a does not affect phase shifting of the circadian clock.

**ASIC1a affects signaling in the pituitary.** To elucidate the mechanisms mediating the ASIC1a regulation of body temperature and voluntary activity, the signaling in the pituitary was further analyzed. RT-qPCR of several pituitary hormones and proteins showed that expression of *Tshb* increased at ZT13 in the pituitary *pars distalis* (PD) of WT but decreased in the PD of ASIC1a⁻/⁻ mice and was higher in WT than in ASIC1a⁻/⁻ mice at ZT13 (Fig. 7a, Supplementary Fig. 5a). The orphan nuclear receptor Rev-Erbα (NR1D1) and the nuclear co-repressor NCOR1 were shown to regulate the circadian rhythm of *Tshb* in the pituitary, with the rhythm of *Ncor1* being opposite of that

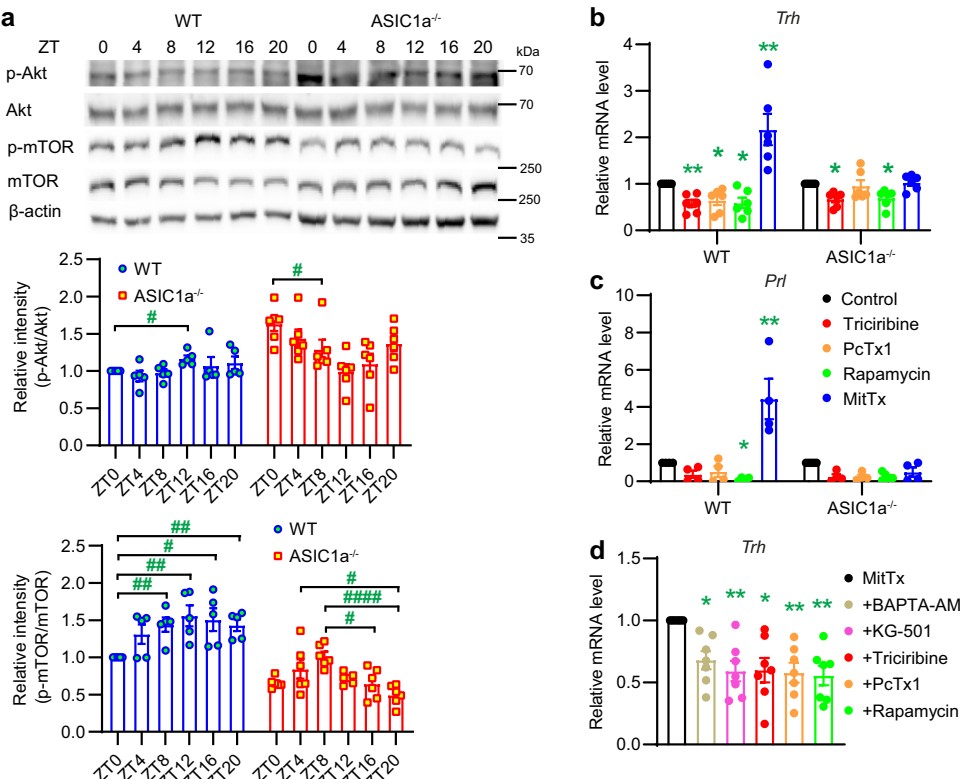

**Fig. 5 Activation of ASIC1a regulates the expression of *Trh* and *Prl*. a** Representative western blots and quantitative analysis of Akt, p-Akt, mTOR, p-mTOR, and β-actin expression are shown for the indicated ZT in WT and ASIC1a$^{-/-}$ mouse hypothalamus. For the quantification of p-Akt/Akt and p-mTOR/mTOR expression ratios, the intensity ratio of each condition was normalized to the WT ratio at ZT0 in each independent experiment, $n = 5$–6 animals per condition. *$p < 0.05$; **$p < 0.01$; ****$p < 0.0001$. Within the same genotype, differences in the p-Akt/Akt or p-mTOR/mTOR expression ratio were compared with each other by one-way ANOVA test and Dunnett's post hoc test, and with the ZT0 WT condition by one-sample $t$ test. **b–d** RT-qPCR analysis of *Trh* (**b**, **d**) and *Prl* (**c**) expression in cultured hypothalamus neurons. **b**, **c** Cultured neurons were exposed to triciribine (10 μM), PcTx1 (10 nM), rapamycin (200 nM), MitTx (2 nM) or vehicle (control) for 2 h, $n = 4$–7 independent experiments. **d** Neurons were pretreated for 30 min with the inhibitors BAPTA-AM (30 μM), KG-501 (10 μM), triciribine (10 μM), PcTx1 (10 nM), or rapamycin (200 nM) and then co-exposed for 2 h with the above inhibitors and 2 nM MitTx. Expression levels measured for each condition are presented relative to the mean of the control (**b**, **c**) or to MitTx-alone (**d**) in the respective experiment; $n = 7$ independent experiments. #$p < 0.05$; ##$p < 0.01$; compared with control of the respective genotype (**b**, **c**) or MitTx (**d**) by one-sample $t$ test. Statistical analysis was done on log2-transformed data. Error bars indicate SEM.

of *Tsh* and *NR1D1*[37]. We found that *Nr1d1* mRNA levels at ZT13 were decreased by ASIC1a deletion in PD, while *Ncor1* mRNA levels were not affected (Fig. S5b, c). The TSH protein level in the PD was measured at ZT17 (time at which ASIC1a$^{-/-}$ mice showed a lower body temperature than WT mice) and at ZT5 (Fig. 7b); there was no difference in TSH abundance between the two time points, nor between genotypes. The expression of *Prl* was decreased at ZT13 relative to ZT1 in the pituitary of WT but not ASIC1a$^{-/-}$ mice (Fig. 7c), opposite to the regulation in the hypothalamus of WT mice, where *Prl* levels were higher at ZT13 than ZT1 (Fig. 4c).

**ASIC1a deletion does not alter thyroid function**. To determine whether the observed changes in the hypothalamus and the pituitary affect thyroid function, we measured the serum concentrations of TSH, T3, and T4 at four time points, ZT1 and ZT13 for comparison with the expression analysis, and ZT5 and ZT17 because the significant difference between the two genotypes in core body temperature was observed around ZT17. This analysis showed a circadian pattern for each of the hormones (Fig. 7d). However, although the amplitude of the circadian fluctuations of TSH and T3 serum levels appeared to be smaller in ASIC1a$^{-/-}$ mice, there were no significant differences between

WT and ASIC1a$^{-/-}$ mice in TSH, T3 or T4 serum levels at any time point. This is consistent with the absence of any difference in TSH protein abundance in the PD between the two genotypes (Fig. 7b). The serum concentrations of T3 and T4 were higher during the inactive period, as reported previously[38,39]. At ZT5 and ZT13, there was a tendency towards higher T4 serum levels in ASIC1a$^{-/-}$ mice compared to WT mice. If the lower body temperature of ASIC1a$^{-/-}$ mice during the active phase were caused by changed signaling along the HPT axis, one would expect lower T4 serum levels in ASIC1a$^{-/-}$ mice during this phase, which was not the case. These data indicate that, despite the effects of ASIC1a on the HPT axis, the lower body temperature in ASIC1a$^{-/-}$ mice is not caused by altered serum levels of thyroid hormones.

In the mediobasal hypothalamus (MBH), local T3 homeostasis is controlled by type II and III iodothyronine deiodinases (DIO2 and DIO3). DIO2 generates T3 from T4, whereas DIO3 degrades T3 to T2 (3,3'-diiodothyronine)[40]. The TSH secreted from the pituitary *pars tuberalis* (PT) binds to the TSH receptor in MBH that activates a cAMP cascade. This TSH-TSH receptor-Dio2 signal pathway in PT/MBH is important for seasonal adaptations; it is not involved in signaling to the thyroid[41]. Interestingly, at ZT13, the TSH concentration in the PT was

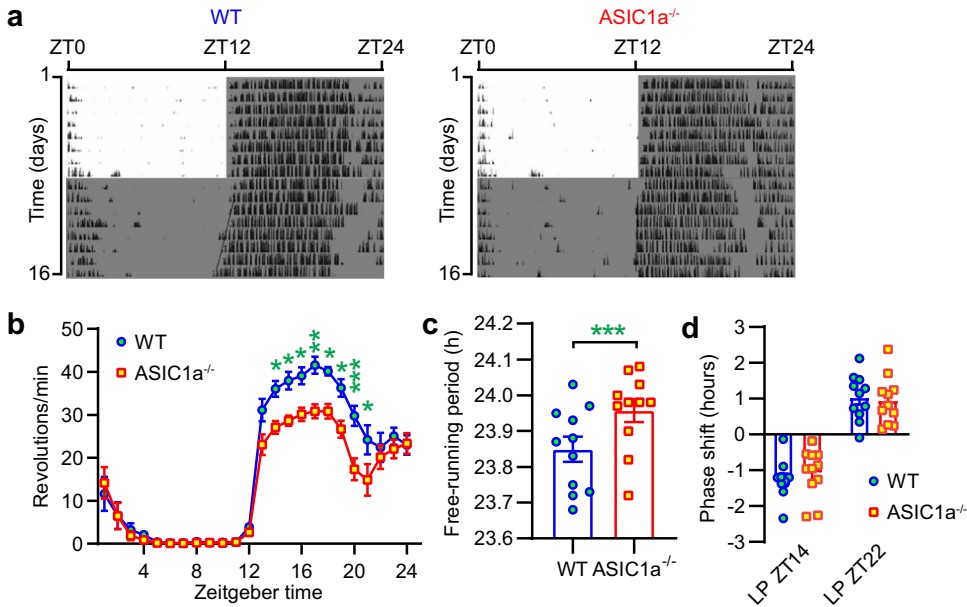

**Fig. 6 ASIC1a regulates circadian rhythm behaviors. a** Representative actograms of wheel-running activity profile of a WT and ASIC1a$^{-/-}$ mouse under 12 h light : 12 h dark (LD) cycle or constant dark (DD, lower half) are shown. Periods of darkness are shown by dark shading. Animals were habituated as previously described[60]. **b** The 7-day average activity of WT and ASIC1a$^{-/-}$ mice under LD conditions (corresponding to days 2–8) was quantified, $n = 11$ WT and 11 ASIC1a$^{-/-}$ mice. *$p < 0.05$; **$p < 0.01$; ***$p < 0.001$, the difference between genotypes, two-way ANOVA repeated measures and Sidak post hoc test. **c** WT and ASIC1a$^{-/-}$ mice were transferred to DD cycle and locomotor activity recordings from days 10 to 16 in free-running conditions were used to assess their internal period length with the $\chi^2$-periodogram analysis, $n = 11$ WT and 11 ASIC1a$^{-/-}$ mice. ***$p < 0.001$, comparison between WT and ASIC1a$^{-/-}$ mice, $t$ test. **d** Phase shifts after light pulses according to the Aschoff type 2 protocol. For each light pulse (ZT14 and ZT22), animals were entrained to an LD cycle for 2 weeks before being released into constant darkness for 10 d. The pulse was administered on the first night after the end of LD, and phase shifts were calculated, $n = 11$ WT and 11 ASIC1a$^{-/-}$ mice (two-way ANOVA). Statistical analysis was done on non-transformed data. Error bars indicate SEM.

strongly increased in ASIC1a$^{-/-}$ mice compared to WT mice (Fig. 7e), although the *Tshb* RNA levels in the PT were not significantly different between ASIC1a$^{-/-}$ and WT mice (Fig. 7f). In WT mice, the expression levels of *Dio2* in the MBH showed a tendency towards lower levels at ZT13 than at ZT1 ($p = 0.16$), while in ASIC1a$^{-/-}$ mice, they were similarly low at both time points (Fig. 7g). These observations may suggest an ASIC1a-dependent T3 formation in the MBH via DIO2, which would inhibit TRH secretion. *Dio3* levels in MBH were not affected by time nor by genotype. Taken together, these data highlight notable changes in the HPT axis of ASIC1a$^{-/-}$ mice, without, however, altered thyroid hormone serum levels that could account for the changes in the core body temperature of these mice. The daily rhythm of pituitary mRNA expression of *Prl*—a hormone known to affect metabolism and activity in mice—is suppressed in ASIC1a$^{-/-}$ mice, pointing to a possible role of PRL in the observed changes.

## Discussion

We show here that the expression of ASIC1a has a daily rhythm in the adult hypothalamus and in embryonic neurosecretory hypothalamic neurons. Global deletion of ASIC1a lowered voluntary activity and the core body temperature during part of the active phase and disrupted the circadian rhythm of the mRNA expression of *Trh* in the hypothalamus, of *Tshb* in the pituitary, and of *Prl* in both hypothalamus and pituitary. Since deletion of ASIC1a did not alter the serum thyroid hormone levels, the lower body temperature in these mice is likely due to a changed metabolism and/or energy expenditure.

TRH from the PVH controls TSH synthesis and release from the pituitary PD (Fig. 7h). We found that the increase in *Trh* mRNA levels in the PVH at night was disrupted in ASIC1a$^{-/-}$

mice. Deletion of ASIC1a ostensibly affected TRH synthesis via the observed changes in electrical signaling in the PVH. The SFR was increased at night relative to the day in WT but not ASIC1a$^{-/-}$ mpd neurons. The SCN sends inhibitory signals to the PVH[42], and modulation of the SFR in the SCN controls circadian behavioral rhythms[21]. The SFR of mouse SCN neurons is higher during the day than at night[43,44]. The observed higher SFR in mpd neurons at night is consistent with a negative regulation by the SCN. Inhibitory inputs to the PVH are mediated by GABAergic signaling[45], and the activation of GABA$_A$ receptors was shown to inhibit ASIC activity in brain neurons[46]. The observed disruption of the circadian changes in mpd neuron SFR of ASIC1a$^{-/-}$ mice indicates that the SFR is positively regulated by ASICs of the PVH or its connected nuclei.

Among the genes upregulated by ASIC1a deletion identified by RNA-seq, several are known to be negatively regulated by Akt and/or mTOR, which suggests that ASIC1a may, via Akt and/or mTOR, inhibit the expression of these genes. In hypothalamic neurons, activation of ASICs is expected to induce an increase in the intracellular Ca$^{2+}$ concentration[4]. Intracellular Ca$^{2+}$ and hypoxia activate PI3K[47]. The inhibition of the ASIC1a-induced increase of *Trh* expression by the Ca$^{2+}$ chelator BAPTA-AM indicates a contribution of the Ca$^{2+}$ entry (Fig. 5d). PI3K acts upstream of Akt[48]. Activation of ASIC1a by MitTx increased the Akt and mTOR activity. In experiments in which ASIC1a had been activated by MitTx, the p-CREB inhibitor KG-501 partially prevented the increase in *Trh* expression, suggesting that mTOR induces *Trh* expression via CREB. Deletion of ASIC1a upregulated in the hypothalamus the p-Akt/Akt and decreased the p-mTOR/mTOR expression ratio (Fig. 5a). The increased p-Akt/Akt expression ratio in ASIC1a$^{-/-}$ mice is opposite from what is expected if ASIC1a activates the Akt-mTOR signaling pathway, suggesting that ASIC1a

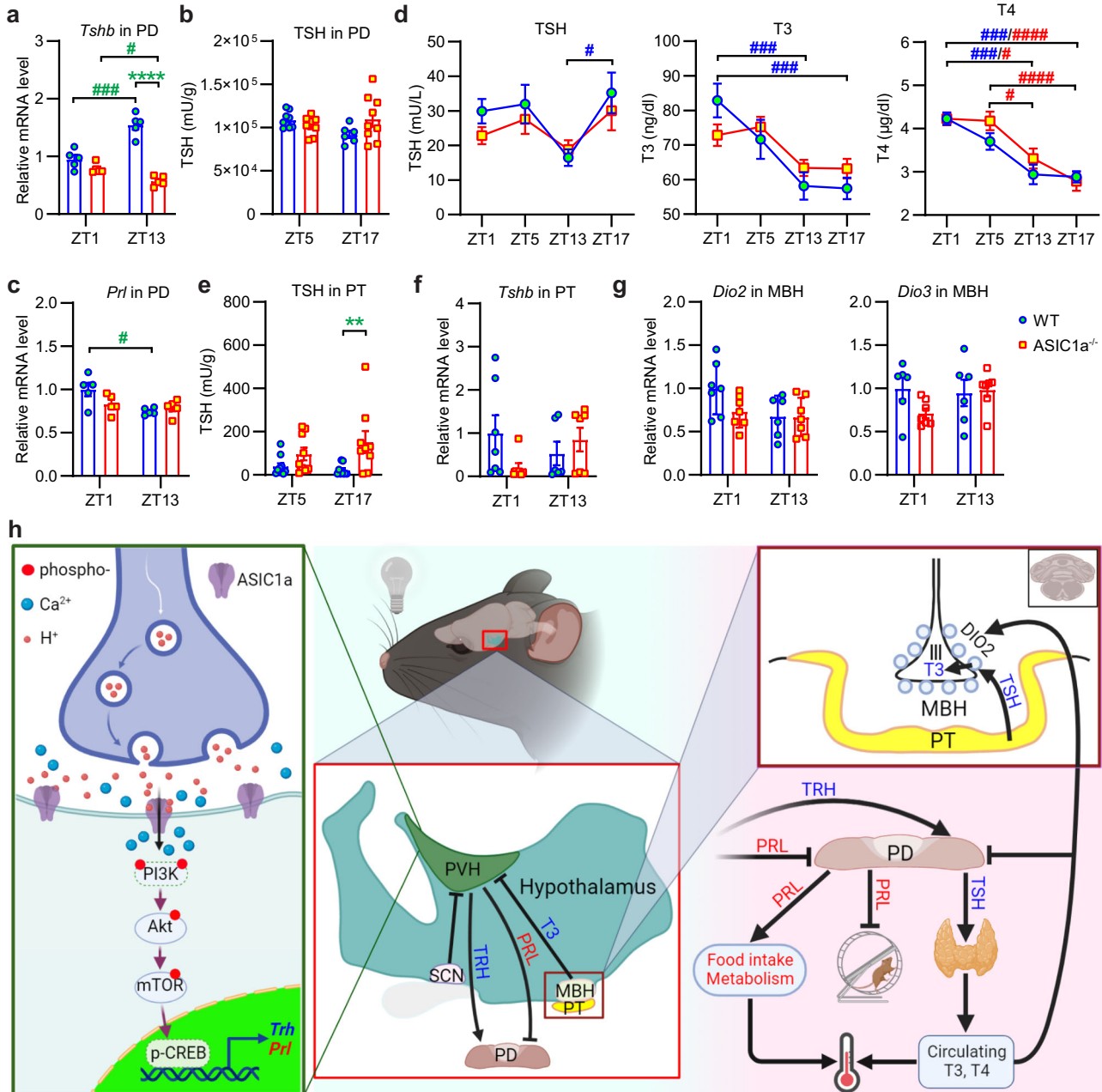

**Fig. 7 Altered signaling in ASIC1a$^{-/-}$ mice at the level of the hypothalamus and pituitary, but not the thyroid. a, c, f, g** mRNA levels quantified by RT-qPCR. **b, d, e** activity or quantity of proteins. In the RT-qPCR experiments, the gene expression levels were normalized to the mean expression of a given gene in WT at ZT1. **a** *Tshb* mRNA levels in pituitary *pars distalis* (PD); $n = 4-5$ animals per condition. **b** Activity of TSH in PD at the indicated time points, expressed as mU TSH/g of protein, $n = 6-9$ animals per condition. **c** *Prl* mRNA levels in the PD of the pituitary, $n = 4-5$ animals per condition. **d** Activity of TSH, T3 and T4 in serum at the indicated time points; $n = 5-9$ animals per condition. **e** Activity of TSH in pituitary *pars tuberalis* (PT) at the indicated time points, expressed as mU TSH/g protein; $n = 6-9$ animals per condition. **f** RT-qPCR analysis of *Tshb* expression in mouse PT; $n = 6-7$ animals per condition. **g** mRNA levels of *Dio2* and *Dio3* in MBH, $n = 6-9$ animals per condition. **p < 0.01; ****, p < 0.0001; WT compared to ASIC1a$^{-/-}$ at the corresponding ZT; #p < 0.05; ###p < 0.001; ####p < 0.0001; comparison for a given genotype between the ZT conditions; two-way ANOVA test and Holm-Sidak's post hoc test. In d, the 2-way ANOVA analysis indicated that the genotype did not add to the variation, while the ZT did (p < 0.05 for TSH, and p < 0.0001 for T3 and T4). Statistical analysis was done on log2-transformed data, except for d where absolute values were used. Error bars indicate SEM. **h** Diagram summarizing the conclusions of the current work (created with BioRender). Left zoom, postsynaptic ASIC1a channels are activated by a drop in the pH in the synaptic cleft in PVH under dark conditions, leading to Ca$^{2+}$ influx. The increase in intracellular Ca$^{2+}$ activates the downstream PI3K/Akt/mTOR/CREB signaling pathway. p-CREB regulates the expression of *Trh* and *Prl*. Center and right, Neuronal signaling in the SCN and PVH controls TRH release. TRH stimulates the release of TSH from pituitary PD. Increased TSH in the PT increases the expression of DIO2, which promotes the secretion of T3 from the MBH to inhibit the expression of Trh in the PVH. Hypothalamic PRL inhibits PRL secretion in the pituitary. The changes in voluntary activity and body temperature may also depend on PRL. In the ASIC1a$^{-/-}$ mice, the serum TSH, T3, and T4 activities were not changed. ASIC1a presumably regulates the body temperature by other changes in the hypothalamus, which include changed activity, food intake, and/or metabolism.

may regulate mTOR activity in hypothalamic neurons via additional pathways that do not involve Akt.

Consistent with the altered *Trh* expression in the hypothalamus, the increase of the *Tshb* expression in the pituitary at ZT13 was suppressed in ASIC1a$^{-/-}$ mice. However, TSH protein abundance in the PD showed no circadian rhythm and was not different between WT and ASIC1a$^{-/-}$ mice. It is not plausible that the increased TSH abundance in the PT of ASIC1a$^{-/-}$ mice could affect the body temperature. TSH from the PT does not stimulate thyroid follicular cells[41], and the measurement of T3 and T4 serum levels at four time points showed no significant differences between WT and ASIC1a$^{-/-}$ mice. Given that the clear differences observed in aspects of HPT signaling do not translate into changes in thyroid hormone serum levels, the regulation of body temperature by ASIC1a must be mediated by other mechanisms. Recent research has highlighted the central actions of thyroid hormones. Thyroid hormones in the arcuate nucleus of the hypothalamus exert an increase in food intake; they also boost, by central actions, the adrenergic tone of the sympathetic nervous system, leading to increased energy expenditure[49,50]. ASIC1a may affect these, or related processes and regulate the body temperature in this way.

A recent study showed that knockdown or pharmacological inhibition of ASIC1a in the PVH of mice caused a decrease in energy expenditure and an increase in food intake, leading to body weight gain, glucose intolerance, and insulin resistance[51], supporting the importance of hypothalamic ASIC1a for the food intake and metabolism. The lower energy expenditure upon ASIC1a knockdown in the PVH is expected to lead to a lower body temperature, as observed in the global ASIC1a$^{-/-}$ mice in the present work. Body temperature was however not measured in the mentioned study[51].

The lower body temperature at ZT17 and ZT18 in ASIC1a$^{-/-}$ compared to WT mice was measured in the period of the day during which ASIC1a$^{-/-}$ mice showed less wheel running (a voluntary activity) than WT mice. The body temperature measurements were done in cages without a running wheel, and spontaneous activity under these conditions, measured with the sensor unit used for the recording of body temperature, was not different between ASIC1a$^{-/-}$ and WT mice. Therefore, the lower body temperature in ASIC$^{-/-}$ mice appears not to be due to decreased activity.

In WT mice, the expression of *Prl* at ZT13 was increased in the hypothalamus, while it was decreased in the pituitary; these daily changes were suppressed by deletion of ASIC1a. PRL is primarily a pituitary hormone, but it is also expressed in several hypothalamic nuclei[24], where local synthesis occurs[52]. PRL of the hypothalamus inhibits, via an increase of dopamine, the basally high PRL secretion by the pituitary in a negative feedback loop[53,54]. There is growing evidence that PRL also affects metabolism outside pregnancy and lactation. Given that PRL promotes food intake[55], the suppressed circadian rhythm in pituitary *Prl* levels in ASIC1a$^{-/-}$ mice might lead to a decreased food intake. Exogenous and endogenous PRL has also been shown to suppress wheel-running activity in mice[56,57]. In the present study, we found that wheel-running activity in the active phase (ZT14-ZT20) was lower in ASIC1a$^{-/-}$ mice than in WT mice. The lower wheel-running activity of ASIC1a$^{-/-}$ mice compared to WT mice at night may be due to the blunted decrease of *Prl* expression in the pituitary at night in ASIC1a$^{-/-}$ animals.

In conclusion, the present study demonstrates that the expression of ASIC1a in the hypothalamus follows a daily rhythm, and that deletion of ASIC1a partially prevents the normal increase in body temperature during the active phase. The lower body temperature in ASIC1a$^{-/-}$ mice is likely due to altered signaling in the hypothalamus that leads to changes in metabolism and/or food intake warranting further characterization.

## Methods

**Animals and ethical approval**. WT and ASIC1a$^{-/-}$ mice were generated by breeding heterozygous mice with a C57BL/6 genetic background. ASIC1a$^{-/-}$ mice[6] were provided by Dr. John Wemmie, University of Iowa. All experiments were carried out with male mice that had access to food and water ad libitum. Male mice were used since there is evidence for a more robust pattern with smaller standard error in some aspects of circadian rhythm, such as wheel running[58]. There are no indications for different effects of this gene deletion between male and female mice. Mice were maintained on a 12 h light/dark cycle unless noted differently. The age of the mice is indicated in the specific Methods sections. The genotype of the mice was determined by PCR analysis of animal tissue[6]. All animal handling procedures were done in accordance with institutional and Swiss guidelines and approved by the veterinary services of the Cantons of Vaud and Fribourg. We have complied with all relevant ethical regulations for animal testing. For isolation of hypothalamus tissue, 3-month-old mice were euthanized by exposure to $CO_2$ at different ZT, decapitated, and the eyes were rapidly removed. The mouse brain was quickly removed and immediately placed on a cold plate. The optic nerve between the olfactory bulbs was carefully cut, and the hypothalamus was removed with curved tweezers. For dissecting the SCN, an ~500 µm thick coronal slice of the hypothalamus was cut with a blade. The slice was placed in a dish filled with cold HBSS, and placed under a dissection microscope, and the SCN was dissected out bilaterally, identified by its position caudal of the optic chiasm[59]. For dissecting the PVH, an ~400 µm thick coronal slice of the hypothalamus was cut with a blade. The slice was placed in a dish filled with cold HBSS, and placed under a dissection microscope, and the PVH was dissected out bilaterally, identified by its position relative to the third ventricle. For dissecting the PT, an ~1 mm thick coronal slice of the hypothalamus was cut with a blade. The slice was placed on a cold plate, freezing the tissue, placed under a dissection microscope, and the PT was dissected out with a blade. For dissecting the PD, the mouse brain was quickly removed, and the PD was taken from the skull.

**Analysis of wheel-running activity**. At 2–3 months of age, the mice were single-housed in custom-built cages with a stainless-steel wire running wheel with a diameter of 11.5 cm, as described previously[60] at $22 \pm 2$ °C. Briefly, the mice had unlimited access to the running wheel in a light- and sound-proof ventilated cabinet, and were entrained to a 12 h dark:12 h light cycle for 10 days prior to the start of experiments. Their running activity under this light/dark (LD) 12:12 cycle was recorded over the next 7 days. *Zeitgeber* time (ZT) 0 corresponds to lights on—and ZT12 corresponds to lights off under the LD 12:12 cycle. On the axis of the wheel, a system with a magnet closed a switch for each wheel revolution. The data was digitized using an interface from Actimetrics and the activity was recorded in 1 min intervals and processed using the ClockLab software version 6.0.54, allowing for analysis of daily activity. Lighting conditions were then changed to constant darkness (DD), which allowed for analysis of mouse activity patterns in the absence of light as a daily *zeitgeber*. Such conditions allow the determination of the free-running period ($\tau$), which is the subjective perception of the internal period length without daily synchronization to light. Free-running periods were determined by both, the $\chi^2$-periodogram and by fitting regression lines through activity onsets. These analyses yielded similar results. In Fig. 6c, the statistical significance is based on the $\chi^2$-periodogram analysis. Mice were re-entrained to an LD 12:12 cycle for 10–14 days in order to analyze resetting of the organism's internal rhythm in response to nocturnal light exposure. The Aschoff type 2 protocol was performed[36]. Briefly, after the re-entrainment, mice were again released into DD. On the first night after the end of LD, a light pulse of 15 min was administered at ZT14 or 22[36]. The animals were then kept in DD for 10 days. Regression lines were fitted through activity onsets before and after the light pulse (the first three days after the pulse were discarded) in order to calculate the phase shift on the first day after the pulse.

**Telemetry device implantation**. Telemetry device implantation was performed in the Cardiovascular Assessment Facility (Centre Hospitalier Universitaire Vaudois) on 12 mice per group. Electrodes (HD-S02, Data Sciences International) were placed in the peritoneal cavity of the anesthetized mice. About 20 min before anesthesia, buprenorphine chlorhydrate (0.1 mg/kg) was injected as analgesia. The mouse was anesthetized by isoflurane inhalation and placed on a warming pad (37–38 °C) for maintenance of body temperature and maintained under anesthesia via a nose cone (1.5–2% in $O_2$, 1 L/min). Ocular gel was applied to hydrate the cornea during the surgical procedure. The skin of the mouse was shaved and disinfected with hydro-alcoholic solution at the level of the abdomen. A small incision was made in the skin and the peritoneal wall was just large enough to allow the transmitter to enter the cavity. The flexible leads were tunneled under the skin towards the right pectoral muscle and the last ribs at about 1 cm of the xiphoid appendix. The peritoneal and skin incisions were then sutured and disinfected. The supply of anesthetic gas was stopped, and the animal was returned to its cage placed on a heating surface until complete wakefulness. After surgery, mice received ibuprofen 20 mg/mL in tap water for 4 days. After 10 days of recovery from surgery, the body temperature and the activity of the mice were measured in the cage in the animal facility. The sensor contained a three-axis accelerometer to report activity measurements, allowing analysis by the Ponemah software (Data Science International). The body temperature and activity were measured in three groups of mice (3–4 mice/group) per genotype for 4 days at an environmental

temperature of 24–25 °C. Average values of the daily cycle were then calculated from the 4-day period.

**Serum hormone measurements**. 36 mice were used in these experiments to collect the serum. Mice used in the experiments were maintained on a 12 h light/dark cycle with food and water *ad libitum*. Blood was obtained by cardiac puncture after $CO_2$ exposure of the mice at ZT1 or ZT13. The blood was then left at room temperature (23 ± 2 °C) for 30 min, before a centrifugation at $2000 \times g$ for 20 min was carried out at room temperature to collect the serum; the serum was immediately stored at −80 °C. Serum TSH, total triiodothyronine (T3), and total thyroxine (T4), were measured at the University of Chicago, as previously described in detail[61]. Briefly, TSH was measured in 40 µL of serum using a sensitive double-antibody precipitation radioimmunoassay (RIA). The limit of serum TSH detection was 10 mU/L. Total $T_4$ concentrations were measured by coated-tube $T_4$ monoclonal RIA from MP Biomedicals (Irvine, CA); the detection limit was 0.5 µg/dL. For $T_3$ measurement, 30 µl serum was first extracted with chloroform-methanol, then dried and constituted in RIA buffer; $T_3$ RIA assay was performed using a specific antibody against $T_3$[62]. The sensitivity was 1 pg per tube.

**Mouse neuron cultures**. 36 pregnant mice and 216 mouse embryos were used in these experiments to obtain cells for hippocampus, hypothalamus, and cortical neuron culture. Hippocampus, hypothalamus, and cortical neuron culture were performed as previously described[18]. Briefly, days 14, 15 pregnant mice were euthanized by exposure to $CO_2$, the embryos were killed, and the hippocampus, cortex, and hypothalamus of the E14-15 embryos were dissected in ice-cold HBSS medium (ThermoFisher). Brain tissues were chopped into small pieces (~1 mm) and incubated at 37 °C for 18 min in 0.05% Trypsin-EDTA (ThermoFisher), then washed three times in Neurobasal medium (ThermoFisher) containing 10% fetal bovine serum (FBS) and dissociated into single cells. After a 5 min centrifugation at 1000 rpm, neurons were re-suspended in Neurobasal/FBS medium. Hypothalamic neurons were seeded at 50,000 cells per 35-mm petri dish. The seeded cells were used for the mRNA assay, and five 10-mm diameter glass coverslips coated with poly-L-lysine were placed into the dishes for functional analysis. For Western blot analysis, cortical neurons were seeded at 150,000 cells per 35-mm petri dish previously coated with poly-L-lysine. The medium was replaced after 12 h by Neurobasal Medium Electro (ThermoFisher) containing the B27 serum-free supplement (ThermoFisher), the GlutaMAX supplement (ThermoFisher), and Gentamicin (10 µg/ml final concentration, ThermoFisher). Neuronal cultures were maintained at 37 °C in a humidified atmosphere of 5% (v/v) $CO_2$ in air, and every 2–3 days half of the medium was replaced with fresh plating medium. Patch-clamp and biochemical assay experiments of neurons were carried out after at least 12 days after seeding. To synchronize the rhythm of cultured neurons, they were exposed at day 11 after seeding for 30 min to 1 µM dexamethasone. Experiments were done at ≥18 h after the synchronization.

**Cultured neuron electrophysiological recording**. Electrophysiological recordings were done using the whole-cell patch-clamp technique in voltage- and current-clamp mode with an EPC10 patch-clamp amplifier (HEKA Elektronik-Harvard Bioscience) as previously described[18]. Solution changes were carried out using computer-controlled electrovalves (cF-8VS) and the MPRE8 perfusion head (Cell MicroControls, Norfolk, VA). Data were acquired with Patchmaster software and analysis of the currents was carried out with Fitmaster (HEKA Elektronik-Harvard Bioscience). The sampling interval and the low-pass filtering were set to 50 µs and to 3 kHz, respectively.

The pipette solution contained, in mM, 120 KCl, 30 NaCl, 10 HEPES, 5 EGTA, 2 MgATP, 1 MgCl₂, and 0.5 CaCl₂, adjusted to pH7.2 with Tris-base. The osmolarity of the pipette solution was 280–300 mOsm (Advanced Instrument Osmometer, Norwood, MA, USA). The extracellular Tyrode solution contained, in mM, 140 NaCl, 5 KCl, 10 glucose, 2 CaCl₂, and 1 MgCl₂, buffered to various pH values with either 10 mM HEPES (pH > 6.0) or 10 mM 2-(N-morpholino)-ethanesulfonic acid (MES; pH ≤ 6.0). The osmolarity of the extracellular solution was 310–320 mOsm. The pH of the solutions was controlled on the day of the experiment and adjusted if necessary. All recordings were performed at room temperature.

**Brain slice preparation and recording of spontaneous firing**. 9-10-week-old mice were deeply anesthetized at ZT1 or ZT13 with isoflurane. The mice were killed by decapitation, and the eyes were rapidly removed. The mouse brain was quickly removed and immediately placed in well-oxygenated (95% O₂/5% CO₂, v/v) ice-cold sucrose-based dissection solution containing (in mM): 110 sucrose, 60 NaCl, 3 KCl, 1.25 NaH₂PO₄, 28 NaHCO₃, 7 MgCl₂, 0.5 CaCl₂, 5 D-glucose. pH was adjusted to 7.4 using NaOH or HCl. Three coronal hypothalamus slices (250 µm thick) containing the paraventricular nucleus (PVH) were obtained using a 7000smz-2 Vibratome (Campden Instruments). Brain slices were incubated at room temperature for at least 1 h in oxygenated artificial cerebrospinal fluid (aCSF) containing (in mM): 120 NaCl, 2.5 KCl, 1 NaH₂PO₄, 26.2 NaHCO₃, 1.3 MgCl₂, 2.5 CaCl₂, 11 D-glucose. pH was adjusted to 7.4 using NaOH or HCl. In the recording chamber, the slices were bathed in oxygenated aCSF (32–34 °C) at a flow

rate of ~2 ml/min. The placement of brain slices was observed using an infrared-differential interference contrast video monitor.

Loose patch-clamp recordings were obtained within 8 h after the slice preparation. The recordings were done using borosilicate glass micropipettes (2–3 MΩ) containing aCSF as the pipette solution. Seals were obtained with gentle or no suction to produce a loose patch seal with a resistance of 10–30 MΩ. Extracellular currents from spontaneous APs were recorded in voltage-clamp mode at 0 mV holding potential. Recordings lasted from 3 min for cells firing regularly at high frequency up to 5 min for cells that were silent or firing at low frequency. Currents were amplified using an Axon 200B amplifier and digitized at 250 kHz using a Digidata 1440 A interface (Molecular Devices). Signals were filtered at 5 kHz and analyzed offline with pCLAMP programs (Axon Instruments).

**Protein extraction and biochemical assay**. 374 mice were used in these experiments to collect the organs. Mice were euthanized by cervical dislocation immediately before the removal of the organs. Mouse tissues were lysed in cold RIPA buffer (20 mM Tris-HCl, 150 mM NaCl, 2 mM EDTA, 1% NP-40 (v/v), 1% sodium deoxycholate (w/v), pH7.5), containing 1:100 diluted protease and phosphatase inhibitor cocktail (Sigma Aldrich), with shaking for 20 min on ice. The mixture was then centrifuged at ~14,000 × g for 15 min at 4 °C, after which the supernatant was collected and the protein concentration was measured using the BCA Protein Assay Kit (23227, ThermoFisher), and all samples were diluted to 0.5 mg protein per mL. The samples in 1× sample loading buffer (0.3 M sucrose, 2% SDS, 2.5 mM EDTA, 60 mM Tris pH8.8, 0.05% (w/v) bromophenol blue, 25 mM DTT) were heated at 95 °C for 10 min.

Western blot analysis was carried out as previously described[63]. Briefly, 10 µl protein samples were separated on 10% SDS-PAGE gels at 100 V, then transferred to 0.2 µM nitrocellulose membranes (Amersham Biosciences) at 4 °C, 100 V for 2 h. After the transfer, the blot was blocked with 5% milk (in TBST buffer, Tris-buffered saline with 0.1% Tween 20 solution) for 1 h at room temperature, followed by 2% BSA in TBST buffer for 1 h at room temperature. The blot was incubated at 4 °C overnight with the primary antibodies, followed, after washing, by the HRP-labeled secondary antibody for 2 h at room temperature. The signals were detected using the Fusion SOLO chemiluminescence system (Vilber Lourmat, Marne-la-Vallée, France) using SuperSignal™ West Femto Maximum Sensitivity Substrate (34095, Thermo Scientific). The following antibodies were used: anti-ASIC1 (1:1000, rabbit, kindly provided by Dr. John Wemmie)[6], anti-actin (1:1000, rabbit; A2066, Sigma Aldrich), anti-BMAL1 (1:1000, rabbit), anti-Akt (1:1000, rabbit; H-136, Santa Cruz), anti-p-Akt (Ser473; 1:1000, mouse; 4051, Cell Signaling), anti-mTOR (1:1000, rabbit; 2972, Cell Signaling), anti-p-mTOR (Ser2448; 1:1000, rabbit; 5536, Cell Signaling), goat anti-mouse IgG (1:2000; B7401, Sigma), and donkey anti-rabbit IgG (1:2000; NA934VS, GE Healthcare). Quantification was done using the ImageJ program. β-actin was used as the total protein control to which the band signals were normalized.

**RNA extraction and gene expression assay**. Mouse tissues were lysed, and RNA was isolated according to the manufacturer's instructions using the RNAqueous™ Total RNA Isolation Kit (AM1931, ThermoFisher) and stored at −80 °C. Briefly, tissues were lysed in lysis solution and were then neutralized, and samples were loaded onto a silica filter for RNA binding on the filter. Samples were then washed thoroughly and eluted with DEPC-treated water. RNA was quantified by Nano-Drop 8000 spectrophotometer (ThermoFisher) and the purity was assessed by the absorbance ratios of 260:280 and 260:230 nm. A ratio of 260:280 ≥ 1.8 and 260:230 between 1.9 and 2.2 was considered acceptable for real-time reverse transcription polymerase chain reaction (RT-qPCR). cDNA was synthesized using the Prime-Script™ RT Master Mix (RR036A, TaKaRa) kit. Real-time PCR analyses were carried out with triplicates of each sample cDNA on QuantStudio 12 K Real-Time PCR System (Applied Biosystems, Inc.) with a SYBR Green Master Mix (4309155, ThermoFisher) under the following conditions: 3 min at 95 °C, followed by 40 cycles of 10 s at 95 °C and 25 s at 60 °C. The primers used are shown in Supplementary Table 1. Gene-specific amplification was confirmed by melt curve analysis. Expression levels were calculated relative to *Gapdh* (brain tissue), *Ppia* (Peptidyl-prolyl isomerase A, pituitary), or *Hprt* (Hypoxanthine-guanine phosphoribosyl-transferase, thyroid gland) based on the $2^{-\Delta\Delta Ct}$ method.

**RNA sequencing**. Genomic RNA was extracted from hypothalamus tissue samples using the Purelink RNA mini kit (12183018 A; Invitrogen, ThermoFisher Scientific) following a standard protocol. RNA quantity was analyzed with the Agilent 5200 Fragment Analyzer System (Agilent Technologies, Inc), using RNA quality number (RQN) as a quality indicator; RQN ≥ 8 was considered acceptable for RNA sequencing. Reverse transcribed RNA was used for sequencing performed at the Lausanne Genomic Technologies Facility (GTF, University of Lausanne). The sequencing library was prepared using the Nextera DNA Library Preparation Kit (Illumina, San Diego, CA, USA) for 100-bp paired-end sequencing runs on Illumina HiSeq 2500, aiming for a 100-fold coverage.

Purity-filtered reads were adapted, and quality trimmed with Cutadapt (v. 1.8). Reads matching to ribosomal RNA sequences were removed with fastq-screen (v. 0.11.1). Remaining reads were further filtered for low complexity with reaper (v. 15-065)[64]. Reads were also aligned to the *Mus musculus* transcriptome

(GRCm38.102) using STAR (v. 2.5.3a) and the estimation of the isoforms abundance was computed using RSEM (v. 1.2.31). The number of read counts per gene locus was summarized with HTSeq-count (v. 0.9.1) using *Mus musculus* genome (GRCm38.102) annotation. The quality of the RNA-seq data alignment was assessed using RSeQC (v. 2.3.7)[65]. Statistical analysis was performed for genes in R (v. 4.0.2). Genes with low counts were filtered out according to the rule of 1 count per million (cpm) in at least 1 sample. Library sizes were scaled using TMM normalization and log-transformed into counts per million or CPM (EdgeR package version 3.28.1).

**Reagents**. All drugs were purchased from Sigma Aldrich (Buchs, Switzerland) unless otherwise mentioned.

**Statistics and reproducibility**. Results are expressed as the mean ± SEM. For qRT-PCR analyses, mRNA levels were normalized to the mean of the control condition. For Western blot analysis and cell culture experiments, values at different time points or different conditions were normalized to the control condition on the same blot, or the same cell culture experiment. For all statistical analyses of normalized data, the log (on the basis of 2) of each data point was calculated and statistical analyses were done with these transformed values. Statistical comparisons were performed using Student's $t$ test for comparison between two groups or for paired comparisons, one-way ANOVA followed by Dunnett's or Tukey post hoc test when more than two groups under same condition were involved, and two-way ANOVA followed by Tukey post hoc test for comparison between two or more groups under two conditions. For Western blot data and experiments that were normalized in the same way as the western blot analyses, the data of the time points except the control (e.g., the control being ZT0, WT in Western blots) were compared among each other with one-way ANOVA followed by Tukey or Dunnett's post hoc test; while the values at different time points were compared to the control value with a one-sample $t$ test.

For data expressed as absolute values, the D'Agostino-Pearson omnibus and Shapiro–Wilk test were used to test for a normal distribution. Statistical tests were applied accordingly, as indicated in the respective figure legends. Statistical tests were carried out with Graphpad Prism 9 (GraphPad, San Diego).

**Reporting summary**. Further information on research design is available in the Nature Portfolio Reporting Summary linked to this article.

## Data availability

RNA-seq data have been deposited at GEO (https://www.ncbi.nlm.nih.gov/geo/query/acc.cgi?acc=GSE185718) and are publicly available. All other experimental data are contained in the manuscript and supplementary files or are available from the corresponding author at reasonable request. Raw data are provided in the file "Supplementary Data 1". Raw images of the blots are provided in Supplementary Fig. 6.

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

## Acknowledgements

We thank Professor Jeffrey G. Tasker (Tulane University) and Dr. Dmitri Firsov (University of Lausanne) for their advice on the project design, and Dr. Olivier Bignucolo and Dr. Frédéric Schutz (Biostatistics Platform, Faculty of Biology and Medicine, University of Lausanne) for help with data analysis. We thank Julien Marquis from the Genomics Technologies Facility, Faculty of Biology and Medicine, University of Lausanne, for carrying out the RNA-seq, Anne Catherine Clerc, Alexandre Sarre, and Corinne Berthonneche of the Cardiovascular Assessment Facility, Faculty of Biology and Medicine, University of Lausanne, for carrying out the body temperature measurement experiments. This research was supported by the Swiss National Science Foundation grants 31003A_172968 to (to S.K.), 310030_184667/1 (to U.A.), 310030_192463 (to R.S.) and 310030_212558 (to G.P.S.) and the National Institutes of Health (USA) grant DK15070 to SR. Zhong Peng was supported by a scholarship grant from the Chinese Scholarship Council.

## Author contributions

Conceptualization, Z.P., S.K.; methodology and investigation, Z.P., P.G.Z., T.M., X.H.L.; supervision: R.S., U.A., G.P.S., S.R., S.K.; writing original draft, Z.P., S.K.; writing, review & editing: Z.P., P.G.Z., T.M., X.H.L., R.S., S.R., U.A., G.P.S., S.K.

## Competing interests

The authors declare no competing interests.

## Additional information



