## [Peer Review File · Communications Biology]

Reviewers' comments:

Reviewer #1 (Remarks to the Author):

This is an intriguing study with a large amount of data reflecting application of a wide variety of biochemical and behavioral techniques. The results suggest that the acid-sensing ion channel 1a (ASIC1a) may be involved in regulating normal diurnal fluctuations in body temperature, via signaling mechanisms within the hypothalamus. Unfortunately, the impact of the work is diminished by a few simple problems in the presentation of the methodology and results.

This is a complex paper, with extensive methodology covering a range of experimental approaches. Some lack of clarity is understandable given the range of techniques involved. The problem, however, is that the omissions and uncertainties in some parts of the methods section undermine confidence in the results. This needs to be corrected. First, a simple omission, in the Results, in several experiments, comparisons are made between hypothalamic tissue samples without the SCN and the SCN itself. I could not find an explanation in the methods section of how the hypothalamus was dissected (boundaries, anatomical landmarks) or how the SCN itself was dissected. Under the section on "Animals and Ethical Approval" it is stated that "WT and ASIC1a^{-/-} mice were generated by breeding of heterozygous mice with a C57BL/6 genetic background." Some rationale should be provided here for only using males in the experiments. Why were females not included? Are the effects of this gene deletion different in females? In addition, there is no information about how the genotypes of the progeny from the breeding colony were confirmed. This was presumably carried out by PCR analysis, but the information should be included in the Methods section.

Perhaps the biggest problem affecting the data is in the Statistics section. It is stated that "Results are expressed as the mean \pm SEM. Statistical comparisons were performed using Student's t-test for comparison between two groups or for paired comparisons, one-way ANOVA followed by Dunnett's or Tukey honesty post-hoc test when more than two groups under same condition were involved, and two-way ANOVA followed by Holm-Sidak's post hoc test for comparison between two or more groups under two conditions." These are all generally accepted procedures for parametric analysis of two group, one or two-factor ANOVA designs. The problem is that the great majority of the results represent ratios standardized (in most cases) to WT controls. A cornerstone of parametric statistics is that data should fit a number of key assumptions, including normalcy, continuous distribution and homogeneity of variance. There is no indication that these assumptions were checked prior to ANOVA. Were the analyses performed on the raw data or on the normalized data? If it was on the normalized data (expressed relative to WT controls) then these assumptions almost certainly were not valid – since the variance for the comparison WT group was artificially compressed relative to the other experimental groups. The usual approach to dealing with such situations is to use non-parametric analysis, which is independent of the data distribution in each treatment group. In some places, there is also inadequate presentation of the statistical results. For example Fig. 6D (Activity of TSH, T3 and T4 in serum at the indicated time points) is interpreted as indicating "there were no significant differences between WT and ASIC1a^{-/-} mice in TSH, T3 or T4 serum levels at any time point." This is based on the lack of differences between the genotypes in the measured hormones at each time point (p. 9 last paragraph) based on the pairwise Holm-Sidak's post-hoc tests. Nowhere are the relevant ANOVA results presented. The pairwise comparisons might not have detected individual genotype differences, but looking at the graphs it seems possible that the patterns of hormonal changes over time (relative to the different ZT times) were in fact different in WT and ASIC1a^{-/-} mice – something that would be apparent in a significant interaction effect in the two-way ANOVA results. It seems unlikely that a more complete statistical analysis would change the basic conclusions of this study. However, errors and incompleteness in reporting of statistical data can have a negative impact on the reader's confidence in the results. The authors are encouraged to seek advice from a qualified statistician, to ensure that these results are interpreted appropriately.

Reviewer #2 (Remarks to the Author):

The manuscript by Peng et al. (COMMSBIO-22-3858) provides data about the daily rhythm of the mouse hypothalamic ASIC1a expression and function. Additionally, in mice lacking ASIC1a gene, the authors registered decrease in the core body temperature and in order to elucidate the underlying mechanisms they performed analysis of the hypothalamic signaling, thyroid function and locomotor activity rhythms. Based on the obtained results, authors came to the several conclusions, including that ASIC1a expression and function displayed daily rhythm and its global deletion lowered the core body temperature during the part of the animal active phase. Furthermore, voluntary activity was decreased and the circadian rhythm of the mRNA expression of Trh in the hypothalamus, Tshb in the pituitary and of Prl in both hypothalamus and pituitary was disrupted.

Although the study offers novel data regarding the circadian rhythm of hypothalamic ASIC1a protein expression and function as well as suggesting the role of this ion channel in the thermoregulatory mechanisms, some issues need to be addressed:

Major comments:

The presentation of results is very complex making it difficult to understand scientific goals and experimental outcomes. Hence I recommend the following:

1. In the method section, the authors should provide the information regarding ambient temperature at which the animals were kept during the experiment.

2. The representative blots of Bmal1 protein expression in the hypothalamus, SCN and hippocampus are shown in the Fig.1. without the presentation of blots quantification on the same figure. The quantification of these blots is presented in the Suppl. Fig. 5a-e. It is not clear why the authors opted for such results separation. In order to review data accordingly and thus interpret results I recommend, combining results that will make the data complete, thus, a new figure that should contain data presented in Fig.1 and parts of data presented in Suppl. Fig. 5 (5a,b,c,d,e). The data presented in the Suppl. Fig. 5f should be added to the Suppl. Fig. 1.

3. Due to the amount of various data presented, Fig. 2. is very difficult to follow. I recommend splitting of data presented in Fig.2 into two new figures, where one would include results regarding ASIC1c-related neuronal activity and the other figure, data related to the body temperature changes (core body temperature, locomotor activity and hypothalamic Trh mRNA expression). Accordingly, corresponding changes should be made in the text.

4. It is not very clear why the authors opted to investigate the circadian rhythm of hypothalamic Akt and mTOR activation. Data presented in Fig 4a and Suppl. Fig. 4a is not relevant for the explanation of ASIC1a-mediated regulation of the expression of Trh and Prl mRNA. The observed variations in circadian pattern of the hypothalamic Akt and mTOR activation can not be related only to the ASIC1a mediated regulation. It is not clear what is the significance of this analysis for the overall results?

5. Furthermore, the authors state that pAkt/Akt ratio was lower in WT compared to ASIC1a^{-/-} hypothalamus (line 209) and than in the following lines (228-229) conclude that results are consistent with an activation of the Akt-mTOR pathway by ASIC1a. Care should be taken in interpreting overall results in this way, considering that the lack of ASIC1a protein resulted in opposite outcomes on the Akt and mTOR activation in the hypothalamus and cortical neurons.

6. The authors state that Trh mRNA expression in the hypothalamus was increased at ZT13 (lines 149-151). Did the authors measure the circadian rhythm of Trh mRNA expression in both genotypes? If so, please provide the circadian pattern of the expression. If not, the authors should give the explanation as to why they chose these time frames (ZT1 and ZT13) to measure Trh mRNA expression?

7. The authors measured expression of BMAIL protein and mRNA in various brain regions of the WT mice. Did the authors measure the BMAIL expression in SCN of the ASIC1a ^{-/-} mice? If so please provide the data.

8. The rationale of the voluntary locomotor activity analysis with respect to ASIC1a role in the core body temperature regulation is not evident considering that the body temperature measurements were not done under these conditions (animal access to the running wheel). Nevertheless, the authors state "The lower body temperature at ZT17 and ZT18 in ASIC1a^{-/-} compared to WT mice correlated with less wheel running (a voluntary activity) of the respective mice at night (line388-389). Please reformulate.

9. The authors should comment on the observed increase in Prl mRNA expression level after incubation of ASIC1a^{-/-} neurons with ASIC1a specific inhibitor (PcTx1)? Would that imply that some other family member has an effect on the hypothalamic PRL expression? (Liu, Y., Hagan, R. & Schoellerman, J. Dual actions of Psalmotoxin at ASIC1a and ASIC2a heteromeric channels (ASIC1a/2a). *Sci Rep* 8, 7179 (2018). <https://doi.org/10.1038/s41598-018-25386-9>)

10. References 26 and 29 are miscited. Please correct.

11. The authors state (lines 384-386): "Given that the clear differences observed in aspects of HPT signaling do not translate into changes in thyroid hormone serum levels, the regulation of body temperature by ASIC1a must be mediated by other mechanisms." Can they speculate on the other, additional mechanisms? How results from the current study relate to the most recently suggested central action of thyroid hormones in thermoregulation? (Rial-Pensado E, Rivas-Limeres V, Grijota-Martínez C, Rodríguez-Díaz A, Capelli V, Barca-Mayo O, Nogueiras R, Mittag J, Diéguez C, López M. Temperature modulates systemic and central actions of thyroid hormones on BAT thermogenesis. *Front Physiol.* 2022 Nov 18;13:1017381. doi: 10.3389/fphys.2022.1017381. PMID: 36467699; PMCID: PMC9716276.; Capelli V, Diéguez C, Mittag J, López M. Thyroid wars: the rise of central actions. *Trends Endocrinol Metab.* 2021 Sep;32(9):659-671. doi: 10.1016/j.tem.2021.05.006. Epub 2021 Jul 19. PMID: 34294513.)

Dear reviewers,

We thank you for reading and analyzing our manuscript, and for your comments, which helped us improve the manuscript. We include in the revised manuscript additional data on the expression of *Trh* mRNA in the hypothalamus and have completely revised the statistical analysis with the help of an expert statistician. This revised statistical analysis affected the significance of only few of the results; it did not affect the major findings of our study. We have added the missing information in the Methods section and have re-organized the Results part based on your recommendations.

Please find below our point-to-point responses to all the points raised.

At the end of this document, we provide a table indicating the changes in the figures.

Sincerely,

Stephan Kellenberger

Reviewer #1 (Remarks to the Author):

This is an intriguing study with a large amount of data reflecting application of a wide variety of biochemical and behavioral techniques. The results suggest that the acid-sensing ion channel 1a (ASIC1a) may be involved in regulating normal diurnal fluctuations in body temperature, via signaling mechanisms within the hypothalamus. Unfortunately, the impact of the work is diminished by a few simple problems in the presentation of the methodology and results.

This is a complex paper, with extensive methodology covering a range of experimental approaches. Some lack of clarity is understandable given the range of techniques involved. The problem, however, is that the omissions and uncertainties in some parts of the methods section undermine confidence in the results. This needs to be corrected.

First, a simple omission, in the Results, in several experiments, comparisons are made between hypothalamic tissue samples without the SCN and the SCN itself. I could not find an explanation in the methods section of how the hypothalamus was dissected (boundaries, anatomical landmarks) or how the SCN itself was dissected.

Response: Information about the dissection is provided in the revised manuscript in the first chapter of the Methods (line 427).

Under the section on "Animals and Ethical Approval" it is stated that "WT and ASIC1a^{-/-} mice were generated by breeding of heterozygous mice with a C57BL/6 genetic background." Some rationale should be provided here for only using males in the experiments. Why were females not included? Are the effects of this gene deletion different in females?

Response: We have carried out this study with males because there are indications that circadian rhythms are more robust in male than female mice. We provide this argument in the Methods of the revised manuscript (line 421): " Male mice were used since there is evidence for a more robust pattern with smaller standard error of some aspects of circadian rhythm, such as wheel running. There are no indications for different effects of this gene deletion between male and female mice."

In addition, there is no information about how the genotypes of the progeny from the breeding colony were confirmed. This was presumably carried out by PCR analysis, but the information should be included in the Methods section.

Response: We indicate in the revised Methods that the genotype was determined by PCR (line 425).

Perhaps the biggest problem affecting the data is in the Statistics section. It is stated that “Results are expressed as the mean \pm SEM. Statistical comparisons were performed using Student's t-test for comparison between two groups or for paired comparisons, one-way ANOVA followed by Dunnett's or Tukey honesty post-hoc test when more than two groups under same condition were involved, and two-way ANOVA followed by Holm-Sidak's post hoc test for comparison between two or more groups under two conditions.” These are all generally accepted procedures for parametric analysis of two group, one or two-factor ANOVA designs. The problem is that the great majority of the results represent ratios standardized (in most cases) to WT controls. A cornerstone of parametric statistics is that data should fit a number of key assumptions, including normalcy, continuous distribution and homogeneity of variance. There is no indication that these assumptions were checked prior to ANOVA. Were the analyses performed on the raw data or on the normalized data? If it was on the normalized data (expressed relative to WT controls) then these assumptions almost certainly were not valid – since the variance for the comparison WT group was artificially compressed relative to the other experimental groups. The usual approach to dealing with such situations is to use non-parametric analysis, which is independent of the data distribution in each treatment group.

Response: As suggested by the reviewer, we have collaborated for the revision with a qualified statistician, Frederic Schütz of our faculty, to make sure that the statistical analyses are correct. For all the normalized data, we have calculated the log (on the basis of 2) of the individual, normalized experimental data, and have carried out the statistical analyses on the log values. This is an accepted way of making sure that the distributions are not distorted. For the statistical analysis of absolute values, we have either also used the log values for statistics, or carried out tests for normal distribution, and based on the outcome and the conditions of the experimental design, did the analysis with ANOVA or with non-parametric tests. This modified statistical analysis resulted in some differences in significance relative to the previous analysis, did however not change the main results or message. The statistical analysis is described in detail in the methods section (Line 642) and in the figure legends.

In some places, there is also inadequate presentation of the statistical results. For example Fig. 6D (Activity of TSH, T3 and T4 in serum at the indicated time points) is interpreted as indicating “there were no significant differences between WT and ASIC1a^{-/-} mice in TSH, T3 or T4 serum levels at any time point.” This is based on the lack of differences between the genotypes in the measured hormones at each time point (p. 9 last paragraph) based on the pairwise Holm-Sidak's post-hoc tests. Nowhere are the relevant ANOVA results presented. The pairwise comparisons might not have detected individual genotype differences, but looking at the graphs it seems possible that the patterns of hormonal changes over time (relative to the different ZT times) were in fact different in WT and ASIC1a^{-/-} mice – something that would be apparent in a significant interaction effect in the two-way ANOVA results.

It seems unlikely that a more complete statistical analysis would change the basic conclusions of this study. However, errors and incompleteness in reporting of statistical data can have a negative impact on the reader's confidence in the results. The authors are encouraged to seek advice from a qualified statistician, to ensure that these results are interpreted appropriately.

Response: We agree that it is important to provide the ANOVA results. The 2-way ANOVA analysis of the Fig. 7d data indicated a contribution of the ZT to the variation, not however of the genotype. This information is provided in the figure legend of the revised manuscript.

Reviewer #2 (Remarks to the Author):

The manuscript by Peng et al. (COMMSBIO-22-3858) provides data about the daily rhythm of the mouse hypothalamic ASIC1a expression and function. Additionally, in mice lacking ASIC1a gene, the authors registered decrease in the core body temperature and in order to elucidate the underlying mechanisms they performed analysis of the hypothalamic signaling, thyroid function and locomotor activity rhythms. Based on the obtained results, authors came to the several conclusions, including that ASIC1a expression and function displayed daily rhythm and its global deletion lowered the core body temperature during the part of the animal active phase. Furthermore, voluntary activity was decreased and the circadian rhythm of the mRNA expression of Trh in the hypothalamus, Tshb in the pituitary and of Prl in both hypothalamus and pituitary was disrupted.

Although the study offers novel data regarding the circadian rhythm of hypothalamic ASIC1a protein expression and function as well as suggesting the role of this ion channel in the thermoregulatory mechanisms, some issues need to be addressed:

Major comments:

The presentation of results is very complex making it difficult to understand scientific goals and experimental outcomes. Hence I recommend the following:

1. In the method section, the authors should provide the information regarding ambient temperature at which the animals were kept during the experiment.

Response: The ambient temperature was 24-25°C in the experiments with the device implantation, and 22±2 °C in the wheel running experiments. This information is provided in the revised manuscript in the methods (line 445 and line 487).

2. The representative blots of Bmal1 protein expression in the hypothalamus, SCN and hippocampus are shown in the Fig.1. without the presentation of blots quantification on the same figure. The quantification of these blots is presented in the Suppl. Fig. 5a-e. It is not clear why the authors opted for such results separation. In order to review data accordingly and thus interpret results I recommend, combining results that will make the data complete, thus, a new figure that should contain data presented in Fig.1 and parts of data presented in Suppl. Fig. 5 (5a,b,c,d,e). The data presented in the Suppl. Fig. 5f should be added to the Suppl. Fig. 1.

Response: According to the suggestions of the reviewer, we have re-organized these figures. In line with the changes in the figures, the BMAL1 expression is presented in the first results

paragraph of the revised manuscript.

3. Due to the amount of various data presented, Fig. 2. is very difficult to follow. I recommend splitting of data presented in Fig.2 into two new figures, where one would include results regarding ASIC1c-related neuronal activity and the other figure, data related to the body temperature changes (core body temperature, locomotor activity and hypothalamic Trh mRNA expression). Accordingly, corresponding changes should be made in the text.

Response: We have split the previous figure 2 into new figures 2 and 3. The part on the hypothalamic Trh mRNA expression has evolved, since we have, based on comment #6 of this reviewer, added the time course of hypothalamic Trh mRNA expression.

4. It is not very clear why the authors opted to investigate the circadian rhythm of hypothalamic Akt and mTOR activation. Data presented in Fig 4a and Suppl. Fig. 4a is not relevant for the explanation of ASIC1a-mediated regulation of the expression of Trh and Prl mRNA. The observed variations in circadian pattern of the hypothalamic Akt and mTOR activation can not be related only to the ASIC1a mediated regulation. It is not clear what is the significance of this analysis for the overall results?

Response: The RNAseq experiments showed that several genes known to be negatively regulated by Akt and/or mTOR are upregulated in ASIC1a^{-/-} hypothalamus. This observation suggested that ASIC1a might control the expression of these genes via Akt and/or mTOR signaling. For this reason, we investigated whether Akt and mTOR signaling was changed. We have adapted the text in the results section to make the motivation for this analysis clearer (line 200).

5. Furthermore, the authors state that pAkt/Akt ratio was lower in WT compared to ASIC1a^{-/-} hypothalamus (line 209) and then in the following lines (228-229) conclude that results are consistent with an activation of the Akt-mTOR pathway by ASIC1a. Care should be taken in interpreting overall results in this way, considering that the lack of ASIC1a protein resulted in opposite outcomes on the Akt and mTOR activation in the hypothalamus and cortical neurons.

Response: We agree that we need to be careful with these statements. We are not completely sure that we have correctly understood the comment regarding opposite outcomes between hypothalamus and cortical neurons, and between the Akt and mTOR pathway. In the experiments with cultured cortical neurons (Fig. S4), we don't compare WT and ASIC1aKO neurons, we rather study the effects of inhibitors or activators on different parameters, as the p-Akt/Akt and p-mTOR/mTOR ratios. Inhibiting ASIC1a in these cultures with PcTx1 did not affect the p-Akt/Akt nor the p-mTOR/mTOR. Such a regulation would be expected if ASIC1a function negatively regulates the mTOR pathway. When interpreting these differences between hypothalamus tissue and cultured cortical neurons, we have to consider that for the hypothalamus tissue, the ASIC1a expression is different between WT and KO animal for the lifetime of the animal, while the cultured neurons were exposed during 2h to PcTx1. In the original submission we interpreted the effect of the genotype on the p-Akt/Akt and the p-mTOR/mTOR ratios in hypothalamus tissue, and the effect of inhibitors on these ratios in cultured cortical neurons separately. In the revised manuscript, we provide a common conclusion/interpretation about these changes after the presentation of the data of Fig. S4 (line 228). We think that in this way it is clearer.

"Both ratios were increased with MitTx in WT but not ASIC1a^{-/-} neurons, consistent with an activation of the Akt-mTOR pathway by ASIC1a; they remained however unaffected by exposure to PcTx1. Cultured cortical neurons were exposed to the drugs during 2h, which is a different condition from the experiments in which Akt/mTOR signaling was compared between hypothalamus tissue of WT and ASIC1a^{-/-} (Fig. 5a). This may explain the absence of an effect of PcTx1 exposure. The observed opposite change of Akt and mTOR activity in hypothalamus tissue by ASIC1a^{-/-} deletion is incompatible with a unique signaling along the Akt-mTOR pathway and suggests that other interacting signaling pathways are also likely affected."

6. The authors state that Trh mRNA expression in the hypothalamus was increased at ZT13 (lines 149-151). Did the authors measure the circadian rhythm of Trh mRNA expression in both genotypes? If so, please provide the circadian pattern of the expression. If not, the authors should give the explanation as to why they chose these time frames (ZT1 and ZT13) to measure Trh mRNA expression?

Response: For the revised manuscript, an analysis of the mRNA expression is provided at 4h intervals, showing that the daily rhythm of Trh mRNA expression is lost in the ASIC1a^{-/-} mice (Figs. 3c-d). We have chosen the time points of ZT1 and ZT13 for the mRNA expression analysis of Trh and of other genes, since this corresponds to the time of 1h after the light-dark change, where we would expect changes in expression to occur. In the context of Trh mRNA expression in the hypothalamus, this corresponds approximately to the time point of the minimal and the maximal levels.

7. The authors measured expression of BMAIL protein and mRNA in various brain regions of the WT mice. Did the authors measure the BMAIL expression in SCN of the ASIC1a^{-/-} mice? If so please provide the data.

Response: We did not measure the daily rhythm of BMAL1 protein nor mRNA expression in the SCN in ASIC1a^{-/-} mice. Therefore, we cannot conclude whether ASIC1a affects the expression of BMAL1 in the SCN.

8. The rationale of the voluntary locomotor activity analysis with respect to ASIC1a role in the core body temperature regulation is not evident considering that the body temperature measurements were not done under these conditions (animal access to the running wheel). Nevertheless, the authors state "The lower body temperature at ZT17 and ZT18 in ASIC1a^{-/-} compared to WT mice correlated with less wheel running (a voluntary activity) of the respective mice at night (line388-389). Please reformulate.

Response: In this paragraph we discuss that the difference in body temperature between WT and KO mice occurred during the period during which, in a separate set of experiments, KO mice showed less voluntary activity than WT mice. We state then that the lower body temperature is not due to less activity, since in the experiments in which the body temperature was measured, the spontaneous activity was not different between WT and KO mice. We agree that in this context, the term "correlate" is misleading. We have replaced this sentence as follows: "The lower body temperature at ZT17 and ZT18 in ASIC1a^{-/-} compared to WT mice was measured in the period of the day during which ASIC1a^{-/-} mice showed less wheel running (a voluntary activity) than WT mice." (line 387).

The rationale for the voluntary locomotor activity analysis was to test whether ASIC1a may be involved in the control of diurnal rhythms (line 262).

9. The authors should comment on the observed increase in Prl mRNA expression level after incubation of ASIC1a^{-/-} neurons with ASIC1a specific inhibitor (PcTx1)? Would that imply that some other family member has an effect on the hypothalamic PRL expression? (Liu, Y., Hagan, R. & Schoellerman, J. Dual actions of Psalmotoxin at ASIC1a and ASIC2a heteromeric channels (ASIC1a/2a). *Sci Rep* 8, 7179 (2018). <https://doi.org/10.1038/s41598-018-25386-9>)

Response: The reviewer refers to the data in Fig. 4c, which is Fig. 5c in the revised manuscript. This figure showed a decrease (not an increase) of Prl mRNA after incubation with PcTx1. With the refined statistical analysis (see response to reviewer #1 comments), this difference is no longer statistically significant. In the rodent brain, the only functional ASICs besides ASIC1a are ASIC2a and ASIC2b. ASIC2a homomeric channels or heteromers with ASIC2b require extremely acidic pH for activation (pH5.0 < pH4.5), and ASIC2b cannot form functional homomeric channels. It is therefore difficult to imagine how such channels could be active in the rodent brain. Since this difference is not statistically significant in the revised manuscript and since a role of ASICs other than ASIC1a in the regulation of Prl expression would be highly hypothetical, we think that it is prudent not to speculate about such a role.

10. References 26 and 29 are miscited. Please correct.

Response: At the place of the previous reference 26 we provide the reference to a review on the Akt-mTOR pathway. It would take too much space to provide a specific reference for each of the genes that is negatively regulated by this pathway. Reference 29 has been replaced.

11. The authors state (lines 384-386): "Given that the clear differences observed in aspects of HPT signaling do not translate into changes in thyroid hormone serum levels, the regulation of body temperature by ASIC1a must be mediated by other mechanisms." Can they speculate on the other, additional mechanisms? How results from the current study relate to the most recently suggested central action of thyroid hormones in thermoregulation? (Rial-Pensado E, Rivas-Limeres V, Grijota-Martínez C, Rodríguez-Díaz A, Capelli V, Barca-Mayo O, Nogueiras R, Mittag J, Diéguez C, López M. Temperature modulates systemic and central actions of thyroid hormones on BAT thermogenesis. *Front Physiol.* 2022 Nov 18;13:1017381. doi: 10.3389/fphys.2022.1017381. PMID: 36467699; PMCID: PMC9716276.; Capelli V, Diéguez C, Mittag J, López M. Thyroid wars: the rise of central actions. *Trends Endocrinol Metab.* 2021 Sep;32(9):659-671. doi: 10.1016/j.tem.2021.05.006. Epub 2021 Jul 19. PMID: 34294513.)

Response: A discussion of these regulations that might be targeted by ASIC1a has been added (line 373).

Changes in Figures

The following general changes were introduced in the figures:

- the original Fig. 1 was re-organized; Fig. 2 was split into 2 figures
- the statistical analysis of all the data was revised. For this reason, there are differences in the statistical significance for some data. In the revised manuscript we

use for all comparisons between genotypes the * symbol, and between time points the # symbol

- new data on the diurnal expression of Trh mRNA in WT and ASIC1a-KO hypothalamus is presented in Fig. 3c-d.

Revised Figure	Contains the following panels of the original figures
1	1a, 1b, S5a, S5b, 1d, 1e, S5d, S5e
2	2a-k
3	2i-j, 2k (+ new c, d)
4	3
5	4
6	5
7	6
S1	1c, S1, S5c, S5f
S2	S2
S3	S3a-c
S4	S4, S3d
S5	S6

REVIEWERS' COMMENTS:

Reviewer #2 (Remarks to the Author):

I thank the authors for addressing all my comments.

Reviewer #3 (Remarks to the Author):

The authors have adequately addressed the concerns of reviewer 1. The statistical analysis section has been much improved. The authors have also added the appropriate statistical details to the figure legends of the revised manuscript and highlighted the contributions of the statistician, Dr. Schutz, in the acknowledgements section. The methodological details have also been clarified and the reorganization of the manuscript make the results easier to follow. There are some grammatical errors in the revised manuscript that need to be corrected eg. the SCN was dissect out - should be 'dissected'. I support the publication of the revised manuscript.

Dear reviewers,

We thank you for reading and analyzing our revised manuscript. The reviewers did not ask for any additional changes. We have therefore only introduced changes asked for by the editor.

Sincerely,
Stephan Kellenberger

Reviewer #2 (Remarks to the Author):

I thank the authors for addressing all my comments.
Response: Thank you for your positive evaluation.

Reviewer #3 (Remarks to the Author):

The authors have adequately addressed the concerns of reviewer 1. The statistical analysis section has been much improved. The authors have also added the appropriate statistical details to the figure legends of the revised manuscript and highlighted the contributions of the statistician, Dr. Schutz, in the acknowledgements section. The methodological details have also been clarified and the reorganization of the manuscript make the results easier to follow. There are some grammatical errors in the revised manuscript that need to be corrected eg. the SCN was dissect out - should be 'dissected'. I support the publication of the revised manuscript.

Response: Thank you for your positive evaluation of our manuscript. We have read the manuscript again and have corrected the errors that we detected.